# Learning Debuggable Models Through Multi-Objective NAS

**Zachariah Carmichael**[*1]  **Tim Moon**[*2]  **Sam Ade Jacobs**[*3]

[1]University of Notre Dame
[2]NVIDIA Corporation
[3]Microsoft

**Abstract**  Monumental advances in deep learning have led to unprecedented achievements across various domains. While the performance of deep neural networks is indubitable, the architectural design and interpretability of such models are nontrivial. Research has been introduced to automate the design of neural network architectures through *neural architecture search* (NAS). Recent progress has made these methods more pragmatic by exploiting distributed computation and novel optimization algorithms. However, there is little work in optimizing architectures for interpretability. To this end, we propose a multi-objective distributed NAS framework that optimizes for both task performance and "introspectability," a surrogate metric for the debuggability of a model. We leverage the non-dominated sorting genetic algorithm (NSGA-II) and explainable AI (XAI) techniques to reward architectures that can be better comprehended by domain experts. The framework is evaluated on several image classification datasets. We demonstrate that jointly optimizing for task error and introspectability leads to more disentangled and debuggable architectures that perform within tolerable error.

## 1 Introduction

The success of deep learning is seemingly ubiquitous in a multitude of domains. A core component of its effectiveness is its ability to automate the feature engineering process. Under this perspective, a natural next step is the automation of the architecture design. To this end, *neural architecture search* (NAS) (Elsken et al., 2019b) has been proposed. Progress in NAS has led to results that supersede the state-of-the-art in several applications, such as image classification (Real et al., 2019) and object detection (Zoph et al., 2018).

While NAS has been effective in automating architecture discovery and topping leaderboards, little attention has been paid to the discovery of interpretable architectures. The automation of interpretability would further minimize the need for "human-in-the-loop" pipelines. Not only does this reduce the manual design needed to meet the constraints of an application, but it also enhances the comprehensibility of the discovered models. Increased comprehensibility aids in model debugging, decreases time to deployment, and instills greater trust.

To this end, we introduce a framework for the joint optimization of task performance and a surrogate for the debuggability of a model. In this work, we put forth the following contributions:

- We develop a new metric to quantify debuggability as the disentanglement between latent representations for different data classes: *introspectability*. We further extend this metric by exploiting hierarchical semantic information from the WordNet database.
- We formalize the discovery of interpretable architectures as a multi-objective optimization problem and adopt an evolutionary approach to NAS that maximizes both accuracy and introspectability by directly optimizing the Pareto frontier. We name our pipeline eXplainable NAS (XNAS).

---

[*]Work done while at Lawrence Livermore National Laboratory.

- We conduct analyses of the accuracy-introspectability trade-off, explore phylogenetic trees to understand the inheritability of objectives, visualize disentangled representations, analyze architectural motifs along the Pareto front, and demonstrate introspectability as a surrogate for trustworthiness and debuggability.

## 2 Background & Related Work

Our work is at the intersection of neural architecture search, explainable AI (XAI), evolutionary algorithms, and multi-objective optimization. We present a brief overview of these topics and discuss related work to put our contributions in context.

**Neural Architecture Search (NAS).** Akin to how deep learning is used to automate feature engineering, NAS algorithms automate *architectural engineering* (Elsken et al., 2019b). NAS algorithms can generally be understood as the composition of three elements: (i) a *search space* that defines the possible neural architectures, (ii) a *search strategy* that explores a search space for candidate solutions, and (iii) a *performance estimation strategy* that determines the fitness of a solution. Of the many approaches to NAS, Bayesian optimization (BO), reinforcement learning (RL), and evolutionary algorithms are the most common. While BO is typically applied to low-dimensional problems, several works have applied it to NAS (Bergstra et al., 2013; Domhan et al., 2015) and it has even surpassed human experts on competition datasets (Mendoza et al., 2016). However, BO has mostly been overshadowed by RL ever since Zoph and Le achieved unprecedented results on NAS benchmarks (Zoph and Le, 2017). The RL problem can be formulated with the evolutionary search space as the agent's action space and the test set error as the reward (Zoph and Le, 2017; Zoph et al., 2018). Alternatively, the RL problem can be posed as a sequential control task (Cai et al., 2018; Wei et al., 2021): given the state of the architecture, what network modification should be applied to improve performance?

While RL-based NAS has achieved state-of-the-art across many benchmarks, it tends to be compute-inefficient and can take thousands of GPU hours to converge (Real et al., 2017; Zoph et al., 2018; Baymurzina et al., 2022). *Neuro-evolutionary* approaches are generally lightweight in comparison, and they notably perform the same as RL approaches on NAS benchmarks (Real et al., 2019; Lopes et al., 2022). The use of evolutionary algorithms for NAS can be traced back decades, e.g. (Miller et al., 1989) uses genetic algorithms to propose architectures that are then trained using backpropagation. While evolutionary algorithms have been used to search for both weights and network architectures (Angeline et al., 1994; Stanley and Miikkulainen, 2002), it is more common to only apply evolution to the architecture and to train the weights with gradient descent (Real et al., 2019, 2017; Elsken et al., 2019a; Baymurzina et al., 2022). Evolutionary algorithms evolve a population of candidate solutions to an optimization problem and each generation is derived from the last by applying mating operations to a set of selected parents. In NAS, an offspring may differ from its parents by an added layer, a changed connection, etc. The quality of solutions is judged by a fitness function and evolution is terminated when a resource or time budget is exceeded.

**Multi-Objective Optimization & NAS.** In a multi-objective optimization problem, there are $m$ objectives $\{f_1, \ldots, f_m\}$, which in the context of NAS may be accuracy, floating point operations (FLOPs), energy, etc. When $m > 1$, it becomes nontrivial to select the optimal solution among the set of all objective vectors $Y = \{\mathbf{y} \in \mathbb{R}^m \mid \mathbf{y} = \{f_1(\mathbf{x}), \ldots, f_m(\mathbf{x})\}\}$ where $\mathbf{x}$ is a candidate solution. There exists a variety of strategies to select solutions, such as optimizing for a weighted sum of the (normalized) objectives, lexicographic sorting, or maintaining Pareto-optimal solutions (Marler and Arora, 2004). We are most interested in the latter approach since it captures the trade-offs between objectives and allows the practitioner to choose the optimal compromise for their use case. The set of Pareto-optimal solutions, also called the Pareto frontier or Pareto front, is the set of non-dominated solutions $\{\mathbf{y}' \in Y \mid \{\mathbf{y} \in Y \mid \mathbf{y} > \mathbf{y}'\} = \emptyset\}$, where $\mathbf{a} > \mathbf{b}$ indicates that $\mathbf{a}$ strictly dominates $\mathbf{b}$, i.e. $|\{f_i(\mathbf{a}) \mid 1 \leq i \leq m, f_i(\mathbf{a}) > f_i(\mathbf{b})\}| = m$.

The non-dominated sorting genetic algorithm-II (NSGA-II) (Deb et al., 2002) is an elitist evolutionary approach to multi-objective optimization. Notably, the authors improve the non-dominated sorting algorithm from cubic to quadratic time complexity. The surviving members of a generation are selected in a binary tournament with preference given to members of the Pareto front. Additional offspring are generated from members in the ranked fronts, i.e. the Pareto fronts computed iteratively after removing the members of the previous front. When a ranked front needs to be subsampled, the crowding distance within the front is used to ensure the full front is represented.

Related to our work, NSGA-Net (Lu et al., 2019) is an evolutionary framework for NAS that employs NSGA-II for multi-objective optimization. Like most evolutionary NAS algorithms, NSGA-Net explores and exploits the search space with a fixed-size population of candidate architectures. The authors demonstrate the effectiveness of population-based NAS and the superiority of NSGA-II over a weighted sum of objectives on the CIFAR-10 and CIFAR-100 datasets. While similar to our framework, our focus is on the design of objectives conducive to interpretability. Furthermore, we scale our method to a distributed cluster and evaluate on more datasets.

Multi-objective optimization of a weighted sum of objectives has been employed by many NAS works (Tan et al., 2019; Hsu et al., 2018). The approach is attractive when the objectives are differentiable since it is amenable to gradient descent by backpropagation. For instance, Multi-Objective NAS (MONAS) (Hsu et al., 2018) uses RL with a weighted combination of accuracy, power, and multiply-accumulate operations (MACs) as the reward. However, there are limitations to optimizing for an aggregate of multiple objectives: it relies on manually tuned coefficients, struggles to accommodate objectives that range over multiple orders of magnitude, and tends to cluster in a small region of the Pareto front.

**Explainable AI (XAI) & NAS.** Some of the intersection between interpretability and NAS has been covered in prior work. In Ru et al. (2021), a NAS framework using the Bayesian optimization search strategy is proposed. For efficiency and interpretability, a Weisfeiler-Lehman graph kernel is used to define a Gaussian process surrogate on the search space, and the gradients are used to identify key motifs that lead to well-performing architectures. Similarly, Adam and Lorraine (2019); Zheng et al. (2022) use alternative techniques to identify key motifs used in the search process. However, their notions of interpretability and disentanglement focus on the search process rather than on the learned models themselves. In this work, we extend NAS to disentangle the latent space of learned models.

## 3 Proposed Framework: XNAS

Following the taxonomy in Elsken et al. (2019b), we break up our method into a search space, search strategy, and performance estimation strategy. We further discuss how we scale the search up to an arbitrary number of compute nodes in Appendix O.

### 3.1 Search Space and Search Strategy

We are interested in exploring complex search spaces beyond simple chains, i.e. multi-branch networks such as ResNet (He et al., 2016) or DenseNet (Huang et al., 2017). To this end, we elect to use the popular NAS-Bench-201 search space (Dong and Yang, 2020), which is comprised of a macro skeleton and a searched cell. A full overview is given in Appendix A.

As we are interested in discovering neural architectures that are both accurate and debuggable, we propose to use multi-objective optimization. We explore and exploit the search space using the Non-Dominated Sorting Genetic Algorithm II (NSGA-II), as introduced in Section 2, with two objectives: accuracy and *introspectability* (introduced in Section 3.2). Because we search for architectures that are both accurate and interpretable, we refer to our approach as eXplainable NAS (XNAS). We generate the initial set of solutions by uniformly sampling the layers of the searched cell (see Appendix A for details). These candidates comprise the first generation of the

*population.* Thereafter, the offspring of the proceeding generation are produced by mating the parents comprising the prior generation. Parents are selected based on the ranked Pareto fronts of the population as described in Section 3.2 and Deb et al. (2002). Because of this selection, there is no notion of a single best solution, but rather a set of non-dominated solutions that characterize the optimal trade-off between all objectives. The details of crossover and mutation are provided in Appendix J.

## 3.2 Performance Evaluation Strategy

We evaluate the performance of an architecture using two objectives: task performance and interpretability (debuggability). The former is simple to define quantitatively as the classification accuracy on the held-out validation split of a dataset. However, interpretability is often treated far more qualitatively and an objective definition eludes community consensus. Furthermore, explaining a model is dependent on the audience, data modality, modeling task, and questions being asked. To disambiguate interpretability in the context of the framework, we state our assumptions: that the user has some technical understanding (e.g. a data scientist or domain expert), that we are interested in understanding the model in classification tasks (e.g. as opposed to the data or the NAS evolution process), and that models that maximize the metric lead to qualitatively discernible trends. To this end, we propose to quantify the interpretability of models as the introspectability of disentangled elements, which we describe in the subsequent subsections. We measure this for supervised classification tasks using the pairwise distances between latent representations of individual classes.

**Introspectability**. Here we formalize the score that we denote as *introspectability*: the degree to which the representations of disparate classes within a neural network $\mathcal{M}$ are disentangled. Let us denote the subset of validation data belonging to class $c$ as $\mathfrak{X}^{(c)} \in \mathbb{R}^{N^{(c)} \times H \times W \times C}$. Given $\mathfrak{X}^{(c)}$ as input to $\mathcal{M}$, denote the activations of layer $l$ as $\Phi^{(c,l)} \in \mathbb{R}^{N^{(c)} \times d_1^{(l)} \times \cdots \times d_n^{(l)}}$. We reshape the activations to have a single dimension of size $d^{(l)} = \prod_{i=1}^n d_i^{(l)}$ such that $\Phi^{(c,l)} \in \mathbb{R}^{N^{(c)} \times d^{(l)}}$. We denote all activations for class $c$ within $\mathcal{M}$ as $\Phi^{(c)} = \|_{l=1}^L \Phi^{(c,l)}$ where $\|$ is the matrix concatenation operator along the columns and $L$ is the number of layers in $\mathcal{M}$. The mean activations for class $c$ are then $\bar{\Phi}^{(c)} = \frac{1}{N^{(c)}} \sum_{i=1}^{N^{(c)}} \Phi_i^{(c)}$ where $|\bar{\Phi}^{(c)}| = \sum_{l=1}^L d^{(l)}$. With these definitions, we then formulate introspectability as (1)

$$\text{Introspectability}(\mathcal{M}, \mathfrak{X}) = \frac{1}{\binom{N_C}{2}} \sum_{c=1}^{N_C} \sum_{k=c+1}^{N_C} D(\bar{\Phi}^{(c)}, \bar{\Phi}^{(k)}) \tag{1}$$

where $D(\cdot, \cdot)$ gives the cosine distance between its two vector arguments and $N_C$ is the number of classes in the classification task.

The motivation for introspectability is to produce trustworthy architectures that have a prediction process that reflects the uncertainty of an instantiated model, a means of probing why decisions were made, and a means of identifying or correcting mispredictions. Architectures that maximize introspectability are easier to debug as the latent representations of each sample are better-calibrated to the confidence of the model. The mean activations per class can be thought of as the centroids that live in latent space, and the distance from each centroid can be thought of as the likelihood that a prediction is correct (with lower values being more likely). This enables us to identify mispredictions and mislabeled data, to identify why mispredictions happen (e.g., due to similar latent representations between classes), and to correct models under-performing on certain classes (e.g., by explicitly emphasizing latent separation between two confounded classes). With its definition, introspectability provides a means for debugging a model and gaining a lens into its prediction process. We run experiments that test these capabilities in Section 4.

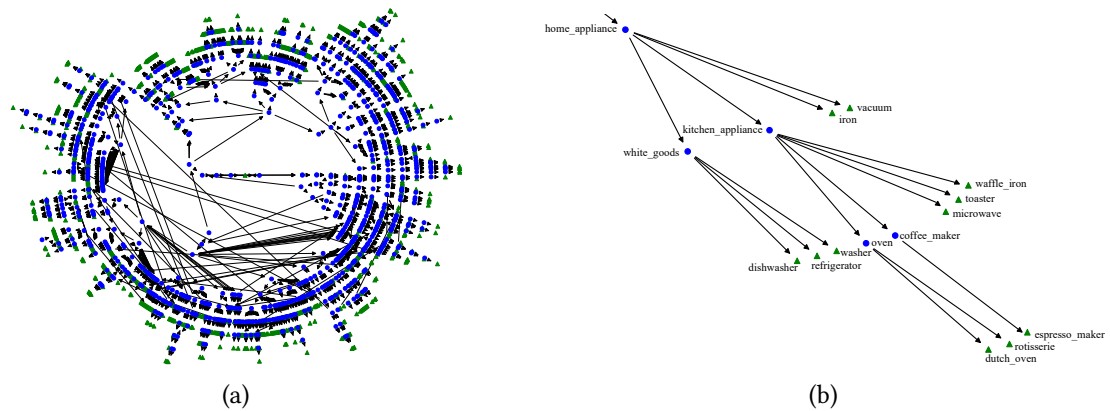

(a)           (b)

Figure 1: (a) The WordNet hyponym-hypernym graph where leaf nodes (green) are the labels of the ImageNet dataset. (b) An inset of the full graph containing the synset "home appliance" and its hyponyms.

When real-values are predicted instead of a class, such as in the regression setting, the target value must be binned into classes and the behavior of models can be interpreted in different regions of the target. For example, these can be "cold," "neutral," and "hot" for weather forecasting, or "low," "medium," and "high" for recidivism risk prediction. These bins can be specified in many ways: uniform splits between the extreme values, based on domain knowledge, clustering (e.g., $k$-means), around higher-risk outcomes, etc. However, we only run experiments on classification tasks in this work.

**Introspectability and WordNet**. We derive a second definition of introspectability based on WordNet (Miller, 1995), a lexical database of the English language. WordNet comprises sets of synonyms (*synsets*) and arises into a hierarchical representation by embedding the transitive relations *hyponyms* (more specific sub-names) and *hypernyms* (more abstract super-names). In computer vision, the labels of the ImageNet database (Deng et al., 2009) are notably derived from WordNet synsets. We visualize all of the labels covered by ImageNet in the hyponym-hypernym graph shown in Figure 1a and 1b. The shortest path distances between two labels in the hyponym-hypernym graph can be used to compute semantic similarity as shown in (2)

$$\texttt{path\_sim}(w_a, w_b) = \frac{1}{\texttt{shortest\_path}(w_a, w_b) + 1} \tag{2}$$

where $w_a$ and $w_b$ are label names. We then weigh the pairwise distances between classes by this similarity as (3)

$$\text{Introspectability}_{\text{WordNet}}(\mathcal{M}, \mathfrak{X}) = \frac{1}{\binom{N_C}{2}} \sum_{c=1}^{N_C} \sum_{k=c+1}^{N_C} D(\bar{\Phi}^{(c)}, \bar{\Phi}^{(k)}) \times S(c, k). \tag{3}$$

The similarity between labels $S$ is given by $S(c, k) = \texttt{path\_sim}(\texttt{name}(c), \texttt{name}(k))$ where $\texttt{name}(\cdot)$ maps the label index to the corresponding label name. Intuitively, the score penalizes models with relatively small distances between dissimilar labels and compensates for small distances between similar labels. To ensure the range of WordNet introspectability is comparable to that of the baseline definition, we normalize the score by dividing by the mean $\texttt{path\_sim}$ value among all pairs of labels in the dataset.

## 4 Experiments & Results

We evaluate XNAS on three image classification datasets: MNIST LeCun et al. (2010), CIFAR-10 Krizhevsky (2009), and ImageNet-16-120 (Dong and Yang, 2020). Thereafter, we conduct analyses

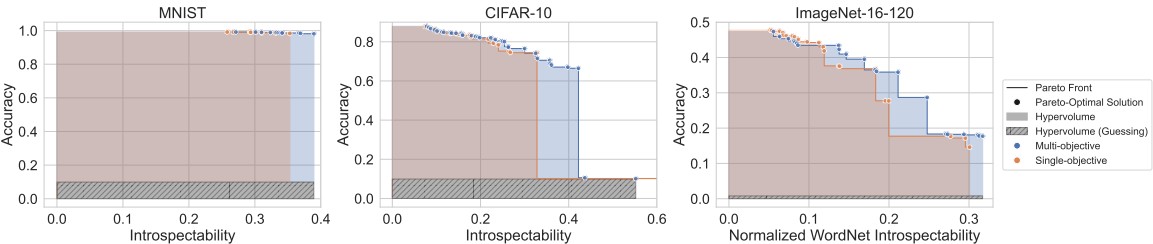

Figure 2: The Pareto fronts and Pareto-optimal solutions are shown for each task. The shaded region visualizes the hypervolume achieved by XNAS. In the first row, the front is visualized for multi-objective XNAS and for single-objective (accuracy only) in the second row. Normalized Introspectability$_{\text{WordNet}}$ is shown for ImageNet-16-120.

to understand the evolution process of XNAS, characterize the Pareto front of each task, and demonstrate the debuggability of higher-introspectability architectures.

### 4.1 Setup and Implementation

The experimental setup and all hyperparameters are detailed in Appendix B. We implement XNAS in Python with code built on the pymoo (Blank and Deb, 2020), DeepHyper (Balaprakash et al., 2018), and Ray (Moritz et al., 2018) libraries. The source code is publicly available at <redacted>.

### 4.2 Metrics

To quantitatively assess the quality of the solutions from a multi-objective search algorithm, we look to *hypervolume* as introduced in Zitzler and Thiele (1998) and improved in Fonseca et al. (2006). Hypervolume is the area of the union of rectangles where each rectangle is formed by a point on the Pareto front and a reference point (such as $(0,0)$). This notion can easily be extended to higher dimensions, i.e. rectangular cuboids are formed with three objectives and hyperrectangles are formed with four or more objectives. In this work, we set the reference point to $(0, 1/N_C)$ where $1/N_C$ is the classification rate of random guessing with balanced data. Note that this reduces the hypervolume range from $[0, 1]$ to $[0, (1-1/N_C)]$. We are interested in setting the reference point here to avoid rewarding models that have not learned the task in any significant capacity.

### 4.3 Results

Table 1 contains the aggregate results for each task and demonstrates the efficacy of using the proposed multi-objective approach. XNAS is compared with the single-objective NAS baseline. While the median population-level accuracy falls slightly compared to single-objective NAS, the maximum accuracy is still comparatively high and there is a substantial increase in hypervolume across experiments. Note that the best accuracy of XNAS on ImageNet-16-120 is on par with the best-performing methods as evaluated in Dong and Yang (2020). We plot the Pareto front of every NAS search result and shade in the hypervolume in Figure 2. The visualizations make clear where the multi-objective approach makes up hypervolume over single-objective (accuracy). Across all tasks, multi-objective covers a larger range of introspectability values. As one would expect, focusing on accuracy tends to cluster the majority of non-dominated solutions in the upper left of the front. The hypervolume of random guessing illustrates why we set the hypervolume reference point to $(0, 1/N_C)$: some solutions manage to achieve high introspectability but are effectively useless since their predictions are no better than random.

As another baseline, introspectability is employed as a *regularization* term. Introspectability is differentiable and in turn, can be used to train models as an auxiliary loss. However, this drastically increases the training computational complexity and memory utilization due to the calculation of pairwise distances and the accumulation of activations. In experiments, this slows down training by

| Dataset | Multi-Objective | Gen. | Max Acc. | Median Acc. | Maximum Intros. | Median Intros. | Hypervolume |
|---|---|---|---|---|---|---|---|
| | | | | Population-Level Statistics | | | |
| MNIST | reg | – | 98.9% | 98.6% | 0.384 | 0.291 | 0.341 |
| | ✗ | 18 | 99.1% | 98.8% | 0.353 | 0.255 | 0.314 |
| | ✓ | 18 | 99.1% | 98.6% | 0.390 | 0.258 | 0.347 |
| | ✓+reg | 18 | 99.0% | 98.6% | 0.503 | 0.303 | **0.424** |
| CIFAR-10 | reg | – | 85.9% | 72.8% | 0.331 | 0.178 | 0.229 |
| | ✗ | 34 | 87.7% | 84.3% | 0.328 | 0.077 | 0.237 |
| | ✓ | 34 | 87.9% | 74.6% | 0.552 | 0.196 | 0.293 |
| | ✓+reg | 34 | 87.7% | 74.2% | 0.654 | 0.249 | **0.361** |
| ImageNet-16-120 | reg | – | 44.8% | 36.2% | 0.318 | 0.087 | 0.099 |
| | ✗ | 11 | 47.8% | 44.9% | 0.301 | 0.053 | 0.099 |
| | ✓ | 11 | 47.3% | 39.1% | 0.317 | 0.104 | 0.111 |
| | ✓+reg | 11 | 47.4% | 39.0% | 0.380 | 0.109 | **0.117** |

Table 1: XNAS experimental results on image classification datasets listing the population-level accuracy and introspectability (Intros.) scores, and hypervolume. Introspectability can be used as a regularizer ("reg"), as described in Section 4.3. We compare to this and single-objective NAS as baselines, as well as to using the multi-objective approach followed by fine-tuning with the regularizer ("✓+reg"). Normalized Introspectability$_{\text{WordNet}}$ is shown for ImageNet-16-120. Multi-objective XNAS with regularization yields the greatest hypervolume across all tasks.

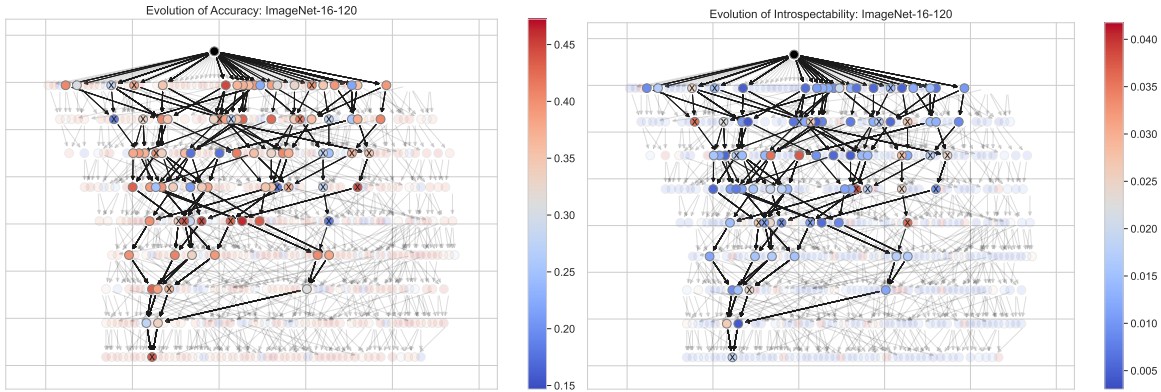

Figure 3: Phylogenetic trees showing the evolution of the population in XNAS on the ImageNet-16-120 task up to the eighth generation. The mating path from the initial population to a solution in the final Pareto front of the search is emphasized. Along the path, parents that were Pareto-optimal within their own generation are marked with an ×. The evolution of accuracy (top) and introspectability (bottom) within the population are shown.

several orders of magnitude. Accordingly, the number of search space evaluations is limited. Regularization achieves introspectability scores competitive with the multi-objective approach on MNIST and ImageNet-16-120, but not CIFAR-10. In addition, the achieved accuracy and hypervolume are hindered due to the reduced evaluations. Synergistically, we also evaluate regularization applied to the Pareto front of the multi-objective approach – XNAS is capable of discovering high-accuracy solutions that are predisposed to higher introspectability via the regularization approach. This combination performs best but demands the additional computation.

To understand the evolution process of XNAS, consider the phylogenetic trees shown in Figure 3. It shows the ancestry for a Pareto-optimal solution from the eighth generation with the ImageNet-16-120 task. While most solutions on a given generation's Pareto front are not directly descended

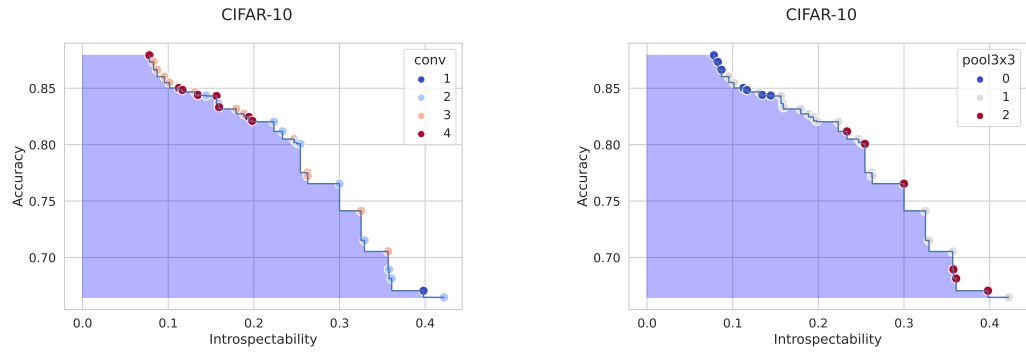

Figure 4: The Pareto front for the CIFAR-10 task with solutions colored by the number of convolutional (left) and pooling (right) layers per cell.

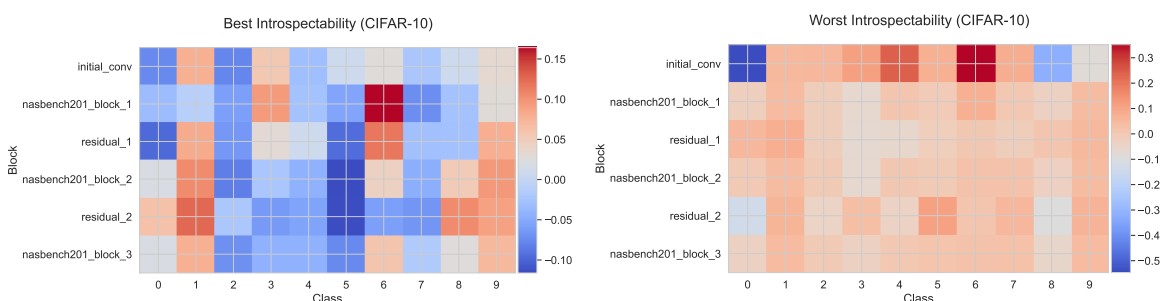

Figure 5: Mean activations heatmap of the model with highest (top) and lowest (bottom) introspectability on the CIFAR-10 task.

from the previous generation's Pareto front, they typically have many Pareto-optimal ancestors. This suggests that Pareto optimality is mildly heritable, although not enough to ensure direct transmission between generations. In addition, the final Pareto-optimal solution had the second-highest cumulative accuracy at its generation. It is not surprising that its ancestors tended to have above-average accuracy and below-average introspectability.

We conduct an analysis of emerging patterns in the architectures discovered across the Pareto front for each task. The methodology for selecting motifs of interest is described in the supplemental material and is accompanied by the corresponding visualizations. In Figure 4, we elect to visualize a pattern that holds for all tasks but is shown for CIFAR-10. Therein, we observe that more accurate models have fewer pooling layers and more convolutional layers, whereas models with greater introspectability exhibit the opposite tendency. These layer types can be seen as one knob that controls the accuracy-introspectability trade-off. Furthermore, a study of the impact of accuracy and introspectability of the generalization error, convergence speed, and the number of parameters is presented in Appendix J. High-introspectability models have lower generalization error, fewer parameters, and faster training times, whereas high-accuracy models exhibit the inverse trend.

To gain a better qualitative understanding of the introspectability metric, we visualize the activations of the Pareto-optimal solutions of each task. In Figure 5, the solutions of the highest and lowest introspectability are shown for CIFAR-10. The MNIST and ImageNet-16-120 activations are shown in the supplemental material. Within each layer, the activations are normalized using z-score normalization. The activations within each block per class are then averaged for the purpose of visualization. The differences between the highest- and lowest-scoring models are quite apparent; the activation patterns for each class in higher-scoring models have notable variance, whereas they are quite constant in lower-scoring models.

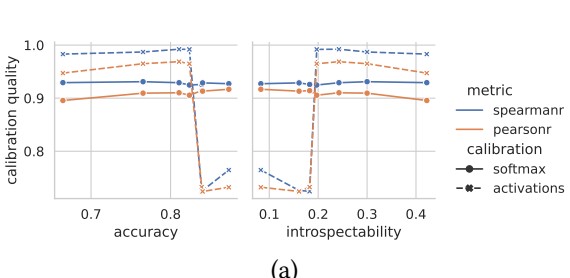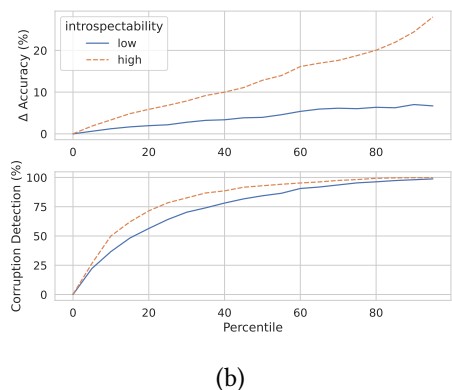

Figure 6: (a) Calibration quality of Pareto-optimal models discovered by XNAS. Calibration using activations improves on softmax for introspectability>0.18. (b) (Top) The change in accuracy due to identified mispredictions as a function of the percentile of sample-wise activation calibration scores. (b) (Bottom) The detection rate of corrupted labels as a function of the percentile of sample-wise activation calibration scores. All results are on CIFAR-10.

**Introspectability, trustworthiness, and debuggability**. Figure 5 demonstrates that increased class-wise disentanglement is a result of optimizing for introspectability. Here, we exploit this to show that introspectability is a surrogate for improved trustworthiness and debuggability of discovered architectures. Typically, *softmax calibration* is employed to assess the confidence and trustworthiness of predictions. This approach interprets the softmax-activated readout layer outputs of DNN classifiers as probabilities. We propose to use an *activations-based calibration* that XNAS indirectly optimizes for. In activations-based calibration, we 1) collect the mean activations per class from training data, 2) collect the activations of the held-out data, 3) compute the cosine distances between the activations of held-out data and the mean activations per class (similar to Eq. (1)), and 4) interpret the distances of the predicted classes as probabilities. To assess calibration quality, we use the Pearson and Spearman rank correlation coefficients of the calibrated probabilities and the corresponding actual accuracy scores at each probability range on a held-out test set[1]. Fig. 6a demonstrates that introspectability estimates calibration quality and outperforms softmax calibration when introspectability> 0.18. Notably, the calibration quality tapers off rapidly at this point, indicating that class-wise latent representations are no longer as disentangled. This is an important observation of this approach – *DNN trustworthiness is a function of class-wise disentanglement.* This experiment serves as motivation that higher-introspectability models are more conducive to trustworthiness and debuggability, as we will demonstrate.

*Identifying mispredictions*   We explore the effect of introspectability on the ability to identify mispredictions. To do so, low-confidence predictions are isolated according to the distribution of activation calibration scores. The quality of identified mispredictions is then quantified as the increase in accuracy due to removing samples. Figure 6b compares misprediction identification between low- and high-introspectability models on CIFAR-10 – high-introspectability models demonstrate greater improvement owing to superior calibration and trustworthiness.

*Debugging data*   We further demonstrate the improved model comprehensibility to debug data. We mislabel samples at a corruption rate of 20% and assess the ability of models to identify the mislabeled data following the distribution of activation calibration scores. Figure 6b shows that high-introspectability models are better equipped to identify bugs in the data on CIFAR-10, achieving a ~20% higher detection rate at the 20th percentile of scored samples.

---

[1]We use 50 linear bins to estimate these quantities.

*Debugging models*  Further model debugging experiments are included in Appendix K – we identify and repair bugs in models based on pairwise activation distances.

## 5 Discussion

As can be observed in Figure 2, there exists a trade-off between accuracy and introspectability, which is more pronounced as the classification task grows more complex, i.e. in the order of MNIST, CIFAR-10, and ImageNet-16-120. We discussed how this trade-off is influenced by the selection of operators within a cell in the previous section. Interestingly, the observation of this phenomenon at first seems to contradict the argument by Rudin et al. that the accuracy-interpretability trade-off is a false dichotomy (Rudin and Radin, 2019). However, an important distinction is that our observation is based on models of the same class and derived from the same search space, rather than on a comparison between different model classes. Moreover, the trade-off is specific to our definition of interpretability and how it is intertwined with performance on the selected datasets.

Our proposed introspectability metric is best suited for technical users, as discussed in Section 3.2, and should not be confused for an all-telling measure of fairness, trust, or reliability. However, the introspectability of models discovered by XNAS has the potential to serve as a requisite criterion before being deployed to users or trusted as a valid model. Furthermore, the metric is designed with simplicity and generality in mind. It can be gamed by an adversary with model-level access, e.g. by inserting futile blocks that are zeroed out and bypassed with a skip connection. This should be addressed in later work, ideally on a per-application basis.

There are several routes for improvement of the XNAS framework. Foremost, the efficiency can be increased by using weight-sharing techniques (Elsken et al., 2019b; Lu et al., 2019), which reduce the evaluation time of offspring. Furthermore, there are uninteresting regions of the Pareto front, depending on the application or end user – NSGA-II can be modified to use reference points of interest to guide the multi-objective search towards more desirable solutions (Deb and Sundar, 2006).

## 6 Broader Impact Statement

Our work aims to bridge the gap between neural architecture search (NAS) and explainable AI (XAI) – by optimizing for both task performance and a surrogate objective for model debuggability, we discover performant architectures that enable some aspects of interpretability. Our framework is a proof of concept of how these two subfields can work together. Several new applications stem naturally. Additional objectives can be utilized from the XAI literature in multi-objective search, even in combination. Existing NAS techniques can be applied to discovering novel architectures conducive to the goals in XAI research. While our approach has potential for broad impact, the limitations of the surrogate metric should be noted. As the text details, the surrogate is not useful to all types of stakeholders nor all aspects of interpretability. Introspectability is particularly useful for data scientists and researchers who understand neural networks at a technical level. The nature of interpretability concerns model trustworthiness, learned generalizations, and understanding the latent space. The surrogate has utility for these stakeholders seeking this type of interpretability. However, potential harm can arise from inappropriate use by users or use cases. As we call for in the text, additional interpretability objectives should be explored in future work for these other settings.

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

# Supplemental Material

## A NAS-Bench-201 Overview

The NAS-Bench-201 search space is comprised of a macro skeleton and a searched cell. An overview is shown in Figure A1.

The first layer of the macro skeleton is a $3 \times 3$ convolutional layer with $F = 16$ filters followed by a batch normalization layer. This is followed by a stack of five searched cells ($F = 16$). A basic residual block ($F = 32$) with a stride of two proceeds the stacked cell block. The shortcut connection is a $2 \times 2$ 2D average pooling layer followed by a $1 \times 1$ convolutional layer. These blocks are alternated, cutting the image dimensions in half and doubling the filters for each set of blocks. The end of the network is a 2D global average pooling layer followed by a fully-connected (dense) classification layer with a softmax activation.

The searched cell can be expressed as a directed acyclic graph where nodes represent data and edges represent operations. The set of operations consists of $3 \times 3$ convolutional blocks, $1 \times 1$ convolutional blocks, $3 \times 3$ average pooling, "zeroize" (equivalent to dropping the edge), and "skip-connect" (equivalent to the identity operator). Note that each convolutional block is comprised of convolution, a rectified linear (ReLU) activation, and batch normalization. All of the convolutions and pooling layers use SAME padding. To prevent cycles, each node is assigned a rank and can only connect to higher-rank nodes. Since there are $V = 4$ nodes in a cell and five operation candidates in the operation set, the total size of the search space is $5^{(\sum_{i=0}^{V-1} i)} = 15,625$ architectures. There are two issues with the search space definition, which the NAS-Bench-201 authors also point out. First, different architecture encodings can result in the same graph. Like the authors, we do not consider isomorphism in the evaluation of architectures[2]. Second, architectures can be disconnected due to the zeroize operation. In this case, the mating operations are reapplied to produce valid offspring.

We represent an architecture in the search space as $\mathbf{x}_i$, a fixed-size list of integers of size $\sum_{i=0}^{V-1} i = 6$ with each element in the range $[1..V]$. Each element of this *encoding* represents (i) a specific operator or operators, such as a convolutional or max pooling layer with specific parameters (e.g. kernel size, strides, etc.), or the lack of an operator (identity) and (ii) how that operator is connected to additional operators in the computational graph.

## B Reproducibility: Experiment Setup and Hyperparameters

**Setup**. We use a cosine annealing Loshchilov and Hutter (2017) learning rate schedule to decay the learning rate from 0.1 to 0 at the end of the last epoch. We also take half an epoch to warm up the learning rate from 0 to 0.1 at midway through the first epoch. Multiple seeds are set for reproducibility – see the code for the NumPy and TensorFlow seed-setting procedure. The seed for each run is stored in each raw result.

**Data preprocessing**. Recall that each raw image $\mathfrak{X}_i$ has a height of $H$ pixels, width of $W$ pixels, and $C$ color channels. We first scale the image by 255 to map the input domain from $[0..255] \subset \mathbb{N}_0$ to $[0, 1] \subset \mathbb{R}$. Then, z-score normalization is applied, i.e. the channel-wise mean of the full dataset $\mathfrak{X}$ is subtracted from each $\mathfrak{X}_i$ and the result of which is divided by the channel-wise standard deviation of $\mathfrak{X}$. The resulting data has channel-wise means of zero and standard deviations of one.

**Data augmentation**. We zero-pad the left and right of each image with $\lceil H/8 \rceil$ pixels and the top and bottom of each image with $\lceil W/8 \rceil$ pixels. Then, each image is randomly cropped following a uniform distribution back to shape $H \times W \times C$. Next, the image is flipped horizontally with a probability of 0.5. The final augmentation applied is cutout Devries and Taylor (2017). Randomly

---

[2]The NAS-Bench-201 authors remark that there are 6,466 architectures with unique topology in the search space due to isomorphisms brought about by the "skip-connect" and "zeroize" operations.

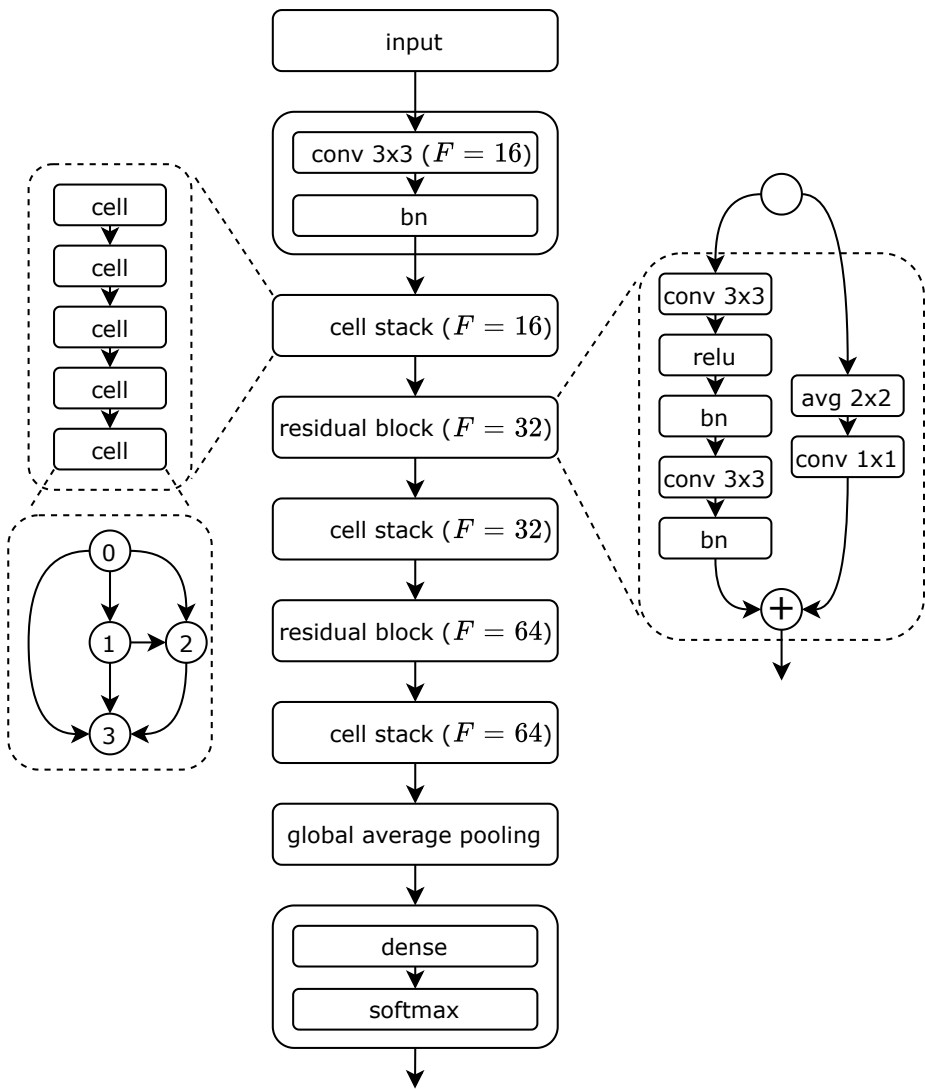

Figure A1: Visualization of the architectures generated by the search space. Note that the first convolutional layer in the main path and the average pooling layer in the shortcut path of each residual block has a stride of 2. conv: 2D convolutional layer. bn: batch normalization. relu: rectified linear (ReLU) activation layer. avg: 2D average pooling layer. $F$: the number of convolutional filters within each layer of a block.

centered rectangular windows with height $2\lceil H/8 \rceil$ and width $2\lceil W/8 \rceil$ are selected to be filled with zeros within the bounds of each image.

We do not allow offspring that have the same architecture as another offspring or a previously evaluated architecture. There are 6 integer variables in the optimization problem, so we set the probability of polynomial mutation per variable to 1/6. Table B1 contains the summary of all hyperparameters used across the experiments.

**ImageNet-16-120**. The ImageNet-16-120 dataset, originally introduced in Chrabaszcz et al. (2017) and adapted by the NAS-Bench-201 benchmark Dong and Yang (2020), is a downsampled version of the ImageNet dataset. The dataset facilitates substantially faster experimentation while permitting satisfactory classification results – performance on ImageNet-16-120 has been shown to be indicative

|  | Hyperparameter | Value |
|---|---|---|
| **Model Training** | Loss | Cross Entropy |
| | Optimizer | SGD |
| | Learning Rate (LR) | 0.1 |
| | LR Schedule | Cosine Decay |
| | Nesterov | Yes |
| | Momentum | 0.9 |
| | Weight Decay | 0.0005 |
| | Batch Size | 512 |
| | Epochs | $5^*$, $12^\dagger$, $200^\ddagger$ |
| | Data Normalization | Z-Score (Channel-Wise) |
| | Data Augmentation | See Text |
| **NSGA-II** | Population Size | 64 |
| | Sampling | Uniform Random |
| | Crossover | Simulated Binary $p = 0.9$, $\eta = 3$ |
| | Mutation | Polynomial $p = 1/6$, $\eta = 3$ |

Table B1: Summary of hyperparameters used across each experiment. $^*$MNIST; $^\dagger$CIFAR-10; $^\ddagger$ImageNet-16-120

of performance across all of ImageNet. Each image in the dataset is resized to $16 \times 16$ pixels and only the data for the first 120 classes are retained.

**Introspectability Regularizer.** Introspectability can be used as a regularization term as it is differentiable. We add this as an auxiliary loss term and naively balance the term with cross entropy with a regularizer weight of 0.5 – this bounds introspectability to the range $[0, 1]$. Because we want to maximize introspectability, we take the cosine similarity instead of the distance. To accumulate activations grouped by classes in TensorFlow, the `tf.scatter_nd` operator is used in implementation. The remaining implementation is straightforward.

## C  Additional Activation Heat Maps

To gain a better qualitative understanding of the introspectability metric, we visualize the activations of the Pareto-optimal solutions of each task. In Figure C2, the solutions of the highest and lowest introspectability are shown for MNIST and ImageNet-16-120 (see main text for CIFAR-10). Within each layer, the activations are normalized using z-score normalization. The activations within each block per class are then averaged for the purpose of visualization. The differences between the highest- and lowest-scoring models are quite apparent; the activation patterns for each class in higher-scoring models have notable variance, whereas they are quite constant in lower-scoring models. The heat maps are best viewed digitally.

## D  Additional PCA Visualizations

The remaining 2D PCA activation visualizations are shown for the MNIST and CIFAR-10 tasks in Figure D1 and Figure D2, respectively. For MNIST, there is little discernible difference between the models with highest and lowest introspectability – this is expected as the difference between these introspectability scores is small (see the main text). For CIFAR-10, an apparent difference between the two models can be observed; the spread of points about the origin is more Gaussian with the higher-scoring model, which, empirically, should indicate a greater mean cosine distance between class representations. It is important to recall that the PCA projection eliminates thousands of dimensions used to represent activations. Naturally, this causes small changes in introspectability to be less apparent in visualizations.

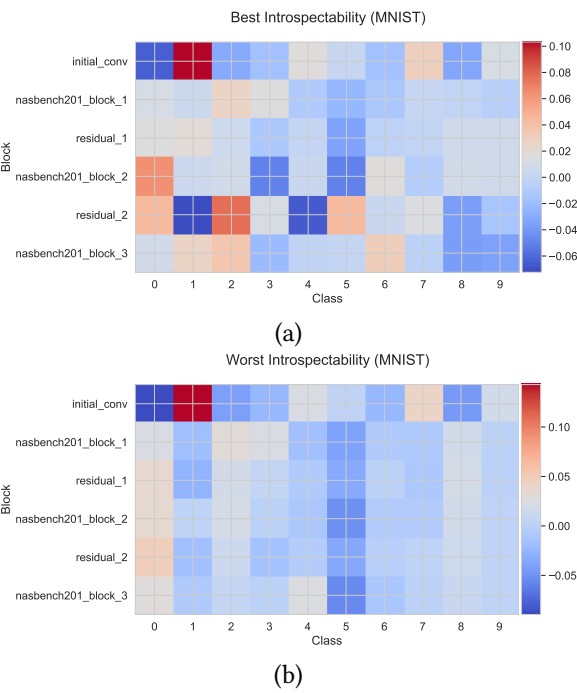

(a)

(b)

Figure C1: Mean activations heatmap of the models with (a) highest and (b) lowest introspectability on the MNIST task.

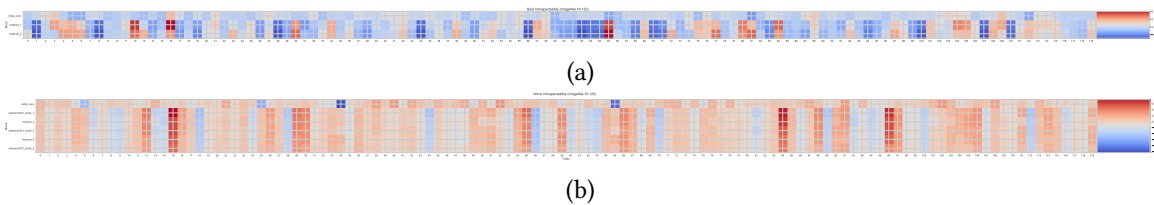

(a)

(b)

Figure C2: Mean activations heatmap of the model with highest introspectability on the ImageNet-16-120 task.

## E  Analysis of Operator Selection

We show the operator-level normalized frequencies selected in the Pareto-optimal solutions of each task in Tables E1-E3. The 3x3 convolutions are most popular across all tasks, followed by either 3x3 average pooling or "zeroize" operators. The skip-connect and 1x1 convolutions are least frequent among these solutions.

| Operation | Normalized Frequency |
|---|---|
| 3x3 Conv2D | 0.51515 |
| 3x3 AvgPool2D | 0.16667 |
| Zeroize | 0.16667 |
| 1x1 Conv2D | 0.09091 |
| Skip-Connect | 0.06061 |

Table E1: Frequency of operations of solutions in the Pareto front (normalized by total cell operations across the Pareto front models) on the MNIST task

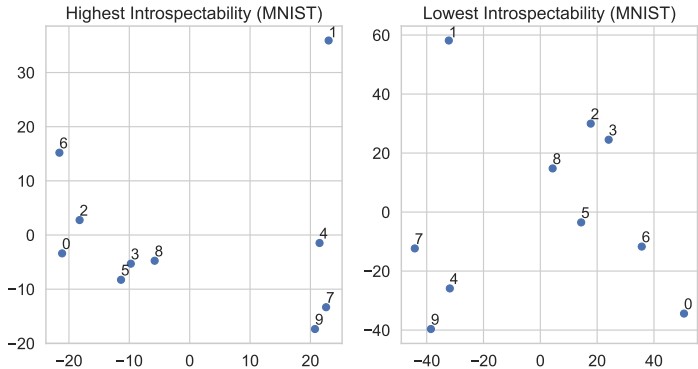

Figure D1: 2D PCA of the mean activations per class from the non-dominated models with highest and lowest introspectability on MNIST.

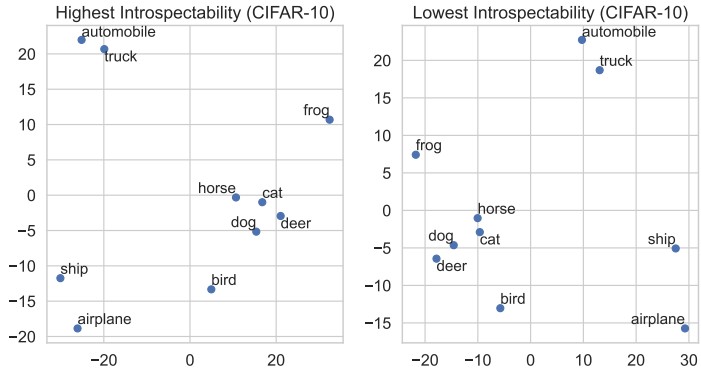

Figure D2: 2D PCA of the mean activations per class from the non-dominated models with highest and lowest introspectability on CIFAR-10.

## F  Frequentist Analysis of Motifs

We conduct analysis of the most common motifs across the Pareto-optimal solutions of each task, as shown in Tables F1-F3. Recall that the integer-coded cells are encoded as follows:

- 0: 3x3 Conv2D
- 1: 1x1 Conv2D
- 2: 3x3 AvgPool2D
- 3: Zeroize
- 4: Skip-Connect

We also use an asterisk (*) to match any operator. Within each table, the encodings of sizes 1 through 5 are shown alongside its normalized frequency of that size. Motifs of size 6 are not shown as we do not evaluate duplicate architectures (other than isomorphisms). The most common motifs reflect the operator frequencies discussed in the previous section. Interestingly, >67% of the Pareto-optimal solutions of each task all have a common motif of size 1, and >45% a common motif of size 2. This suggests that certain cell topologies exhibit inductive biases specific to the task.

## G  Comparing Motifs Across the Pareto Front

**Motif Discovery**.

| Operation | Normalized Frequency |
|---|---|
| 3x3 Conv2D | 0.44444 |
| Zeroize | 0.24306 |
| 3x3 AvgPool2D | 0.19097 |
| Skip-Connect | 0.06250 |
| 1x1 Conv2D | 0.05903 |

Table E2: Frequency of operations of solutions in the Pareto front (normalized by total cell operations across the Pareto front models) on the CIFAR-10 task

| Operation | Normalized Frequency |
|---|---|
| 3x3 Conv2D | 0.48039 |
| 3x3 AvgPool2D | 0.21078 |
| Zeroize | 0.10784 |
| Skip-Connect | 0.10784 |
| 1x1 Conv2D | 0.09314 |

Table E3: Frequency of operations of solutions in the Pareto front (normalized by total cell operations across the Pareto front models) on the ImageNet-16-120 task

1. Assemble the following data into a tabular structure: architecture encoding, accuracy, and introspectability for the Pareto front of the solutions

2. Sort the data by accuracy and then introspectability which results in data with ascending accuracy and descending introspectability

3. Record the count of each block for each architecture encoding

4. For each architecture encoding in the sorted data, enumerate all applicable motifs of size 1 to 5 (motifs of size 6 cannot exist as architectures are not evaluated multiple times). For example, some architecture encoding $[a, b, c, d, e, f]$ has $\sum_{k=1}^{5} \binom{6}{k}$ motifs, e.g. $[a, *, *, *, e, *]$ and $[*, b, c, *, *, f]$ but not, say, $[*, b, c, *, *, e]$. This is nearly equivalent to its power set minus $\emptyset$ and the original sequence. An asterisk here implies a match with any other operator, and thus allows for the comparison of motifs between different architectures

5. For each motif, compute the absolute value of the Spearman rank correlation coefficient between the ranks of the solutions in the sorted data and whether each solution has the motif. If applicable, the count of the operator is used instead of a simple indicator flag. The intuition here is that we discover interesting architectures that demonstrably are favored more in one part of the Pareto front than another, e.g. the high-accuracy vs. high-introspectability regions

6. In addition to each motif having a correlation score, we also record the support (the number of solutions with the motif) and the motif size

7. Compute the Pareto front of the scored motifs (the costs being the correlation score, the support and the motif size) to identify the most salient motifs. We heuristically eliminate motifs that have support less than 3 or correlation less than 0.2

All figures from this discovery are present at the end of the appendices for the sake of space.

| Size | Normalized Frequency | Encoding |
|------|----------------------|----------|
| 1 | 0.81818 | [0 * * * * *] |
| 2 | 0.45455 | [0 * * * 3 *] |
| 2 | 0.45455 | [0 * 0 * * *] |
| 2 | 0.45455 | [0 * * * * 0] |
| 2 | 0.45455 | [* * 0 * * 0] |
| 3 | 0.36364 | [0 * 0 * * 0] |
| 4 | 0.27273 | [0 * 0 4 * 0] |
| 5 | 0.18182 | [0 0 0 4 * 0] |
| 5 | 0.18182 | [0 * 0 4 3 0] |

Table F1: Frequency of encodings of solutions in the Pareto front (normalized by the number of Pareto-optimal solutions) on the MNIST task. The top motif (motifs if tied frequency) for each size is shown only

| Size | Normalized Frequency | Encoding |
|------|----------------------|----------|
| 1 | 0.70833 | [* * * * 3 *] |
| 2 | 0.50000 | [* * * * 3 0] |
| 3 | 0.29167 | [0 * 0 * 3 *] |
| 3 | 0.29167 | [* * 0 * 3 0] |
| 4 | 0.18750 | [0 * 0 * 3 0] |
| 5 | 0.08333 | [0 * 0 1 3 0] |

Table F2: Frequency of encodings of solutions in the Pareto front (normalized by the number of Pareto-optimal solutions) on the CIFAR-10 task. The top motif (motifs if tied frequency) for each size is shown only

## H  Comparing Evolution of Single- and Multi-Objective Search

We illustrate the evolution of accuracy and introspectability of the models on the Pareto front over each generation in Figure H1-Figure H3. Each figure contrasts single-objective with multi-objective optimization to better understand the benefit of NSGA-II in our framework. Note that we do not expect a strict increase in each objective at each generation, which would be expected for population-level statistics, as opposed to statistics within the Pareto front. With single-objective optimization, we can observe that solutions with higher introspectability tend to lie beyond the 95% confidence interval. This indicates fluke solutions, whereas multi-objective more confidently produces higher-introspectability solutions.

## I  Comparing XNAS Accuracy with Related NAS Methods

We compare XNAS to other multi-objective approaches on the CIFAR-10 task. Building on the collected results and approach from Lu et al. (2019), we take the architecture with the best accuracy and increase the number of filters by a factor of four. We then perform full training on the CIFAR-10 dataset for 200 epochs. The comparison of results and methods is shown in Table I1. While XNAS does not achieve the best accuracy (nor was this the objective of this research), the result is still competitive, especially considering the trade-off between accuracy and introspectability.

| Size | Normalized Frequency | Encoding |
|---|---|---|
| 1 | 0.67647 | [* 0 * * * *] |
| 2 | 0.47059 | [* 0 * * * 0] |
| 3 | 0.23529 | [* 0 * 0 * 0] |
| 4 | 0.11765 | [2 0 * 1 * 0] |
| 4 | 0.11765 | [* 0 * 0 0 0] |
| 4 | 0.11765 | [0 0 * * 0 0] |
| 4 | 0.11765 | [0 0 * 0 * 0] |
| 4 | 0.11765 | [0 * * 0 0 0] |
| 5 | 0.05882 | [3 0 2 2 * 3] |
| 5 | 0.05882 | [2 0 1 1 * 0] |
| 5 | 0.05882 | [2 0 * 1 4 0] |
| 5 | 0.05882 | [2 0 4 0 0 *] |
| 5 | 0.05882 | [2 0 * 0 0 0] |

Table F3: Frequency of encodings of solutions in the Pareto front (normalized by the number of Pareto-optimal solutions) on the ImageNet-16-120 task. The top motif (motifs if tied frequency, up to 5) for each size is shown only

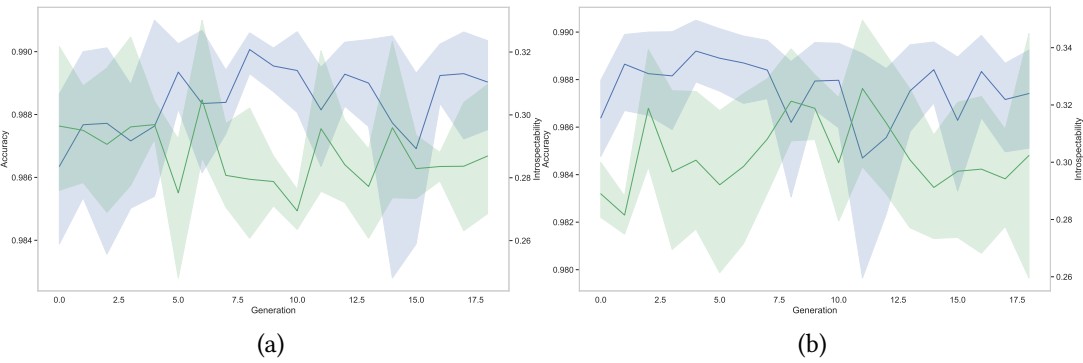

(a)                                        (b)

Figure H1: The mean accuracy (blue) and introspectability (green) of the Pareto front solutions each generation on the MNIST task. (a) single-objective; (b) multi-objective. The shaded region indicates the 95% confidence interval of solutions at each generation.

## J  Additional Ablation Studies

We perform additional studies to understand the relationships between the objectives, accuracy and introspectability, and the generalization error, number of parameters, and training speed of architectures. Figure J1 demonstrates that introspectability and accuracy have an inverse relationship on the generalization error – this error increases with high-accuracy models and decreases with high-introspectability models. Likewise, the trend can be observed with the number of parameters and training speed as shown in Figures J2 and J3, respectively. These figures follow a similar trend as the number of parameters correlates with the number of FLOPs and thus the training time. As discussed in Section 4.4 of the main text, high-introspectability networks tend to have a more pooling layers whereas high-accuracy networks have more convolutional layers. This helps to explain the trends observed in the number of parameters. A takeaway from this analysis is that the trade-off between accuracy and introspectability also implies a trade-off in parameters (and FLOPs), training time, and generalization error.

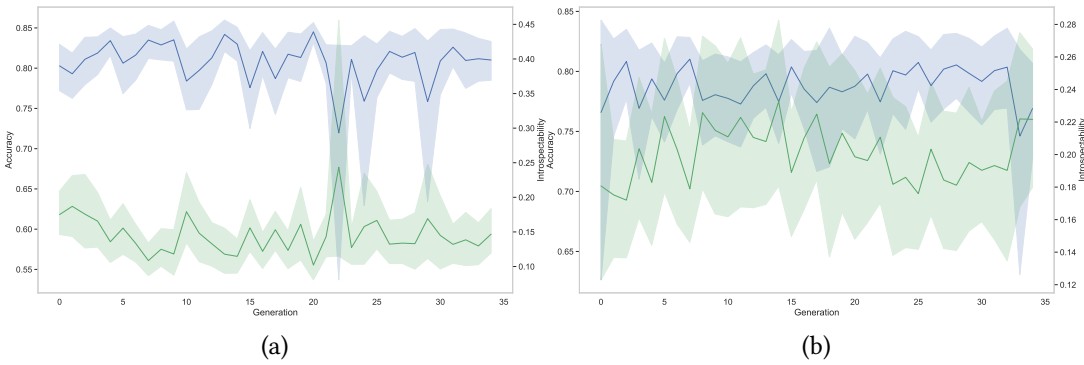

Figure H2: The mean accuracy (blue) and introspectability (green) of the Pareto front solutions each generation on the CIFAR-10 task. (a) single-objective; (b) multi-objective. The shaded region indicates the 95% confidence interval of solutions at each generation.

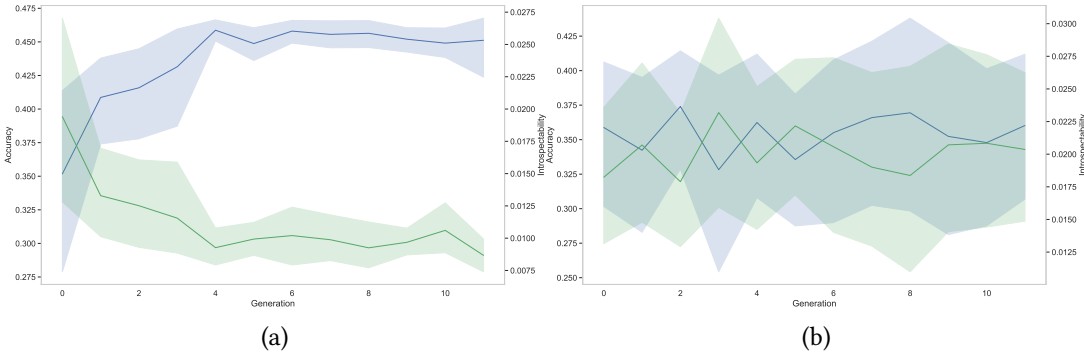

Figure H3: The mean accuracy (blue) and introspectability (green) of the Pareto front solutions each generation on the ImageNet-16-120 task. (a) single-objective; (b) multi-objective. The shaded region indicates the 95% confidence interval of solutions at each generation.

## K  Model Debugging Experiments

Here, we study the ability of our activations calibration approach to correct bugs in models. We first demonstrate that there is a strong connection between the pairwise activation distances used in the formulation of introspectability and the ground truth confusion matrix. To make this comparison, the pairwise distances are negated as disentanglement (separation) between class representations is posited to correlate with confounding. Since the distance between the activations of a class and itself is 0, ideally, the distance between such and the activations of other classes is maximized. There is no information about ground truth available in computing pairwise distances, i.e. the computation is symmetrical and unconditioned. In turn, we compare this information to a confusion matrix folded along the diagonal. This means that element $x_{i,j}, i \neq j$ in the folded confusion matrix is equivalent to the sum $x_{i,j} + x_{j,i}$ in the original confusion matrix. On a higher level, each element $x_{i,j}$ is either the number of true positives for a class (when considering the diagonal), or the support of class $i$ being predicted when class $j$ were true and the support of the converse. To support this, we measure the correlation between the negated pairwise activation distances and the folded ground truth confusion matrix across Pareto optimal models trained on CIFAR-10. High-introspectability models achieve a correlation of $\rho = 0.85$ while low-introspectability models achieve a correlation of $\rho = 0.59$. With this motivation, we demonstrate how model bugs can be identified and corrected in the following case study.

| Method | Error | Other Objective | Compute |
|--------|-------|-----------------|---------|
| PPP-Net Dong et al. (2018) | 4.36% | FLOPs, # parameters, or inference time | Nvidia Titan X |
| MONAS Hsu et al. (2018) | 4.34% | Power | Nvidia 1080 Ti |
| NSGA-Net Lu et al. (2019) | 3.85% | FLOPs | Nvidia 1080 Ti 8 GPU Days |
| XNAS | 4.45% | Introspectability | Nvidia Tesla P100 6 GPU Days |

Table I1: Multi-objective methods for CIFAR-10 (best accuracy for each method). Table adapted from Lu et al. (2019)

**Case Study: Bug Identification and Correction.** Figure K1 demonstrates for a random higher-introspectability model trained on CIFAR-10 that there is strong correlation between the negated pairwise activation distances and the ground truth confusion matrix folded along the diagonal ($\rho = 0.81$). Noticeably, the model confounds the classes 3 and 5, which is reflected in the pairwise activations as the smallest distance (largest negated distance).

With the bug in the model identified, we formulate a strategy to mitigate the issue. The key of our approach is to push the representations of classes 3 and 5 apart in order to reduce the confounding of one another. We accomplish this by using the introspectability regularizer approach with pairwise coefficients. The generalization of this to arbitrary pairs is formalized in Eq. (4).

$$\text{Introspectability}_{\text{reg}}(\mathcal{M}, \mathfrak{X}) = \frac{-1}{\binom{N_C}{2}} \sum_{c=1}^{N_C} \sum_{k=c+1}^{N_C} D(\bar{\Phi}^{(c)}, \bar{\Phi}^{(k)}) \times \omega_{i,j} \qquad (4)$$

where $\omega_{i,j}$ is a weight for each class pair $(i, j)$. With every $\omega_{i,j} = 1$ this is equivalent to the untargeted introspectability regularizer. If the aim is to target all confounded predictions, one can set all $\omega_{i,j}$ proportionally to the pairwise activation distances (or folded confusion matrix). However, we target a single pair in this case study. In our experiment, the model is trained with the regularization term for an additional 5 epochs, a learning rate of 0.001, $\omega_{3,5} = 25$, and all other $\omega_{i,j} = 1$. The results are visualized in Figure K2. The approach is stronger in identifying bugs than mitigating them, although there is improvement without significant degradation of accuracy ($\pm 0.6\%$ across 10 trials). We leave the tuning of hyperparameters and alternative weighting schemes to future exploration.

## L  Extended Background

**DNN Inspection within Explainable AI (XAI).** The opaque nature of deep neural networks (DNNs) has ultimately led to the sub-field of explainable AI (XAI) Gunning (2019), which was denominated in 2016 by DARPA, although relevant work predates this by years. Relevant to the subject matter of this work are XAI methods of DNN inspection. This suite of methods enables the debugging of model behavior, the detection of dataset errors, and the development of adversarial attacks. The authors of Koh and Liang (2017) scale influence functions, a robust statistics method, to DNNs to understand the effect of training points on a prediction. DNN visualization tools have been proposed to provide qualitative modes of analysis. Notably, Erhan et al. (2009); Yosinski et al. (2015) provide tools for visualizations by gradient ascent, deconvolution for highlighting input images, and discovering preferred input patterns for each class. Probing-based methods aim to qualify the role of DNN internal elements (neurons, latent representations, etc.). In Kim et al. (2018); Bau

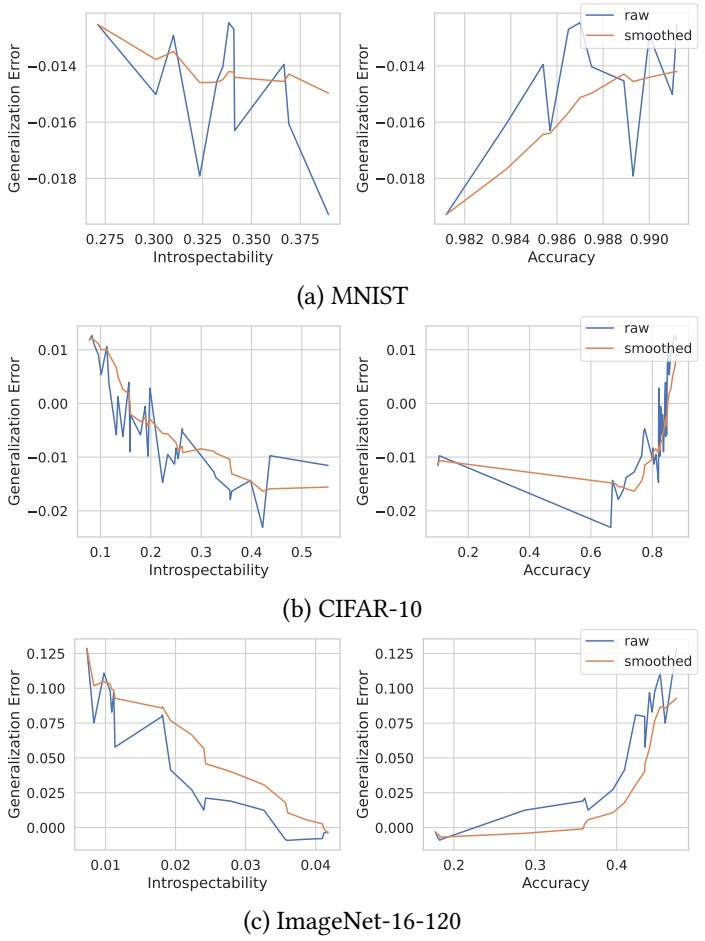

(a) MNIST

(b) CIFAR-10

(c) ImageNet-16-120

Figure J1: The effect of introspectability and accuracy on generalization error across the tasks.

et al. (2020), methods are proposed to relate DNN internals to semantic concepts, such as textures, shapes, colors, or even people. Another approach introduced in Ghorbani and Zou (2020) is to use Shapley values from game theory to quantify the influence each neuron has on overall DNN error.

More related to our work are those related to disentanglement, i.e. the separation of concept- or class-relevant information in a network. For instance, Zhang et al. (2018) proposes the learning of interpretable CNN filters by coercing feature maps to resemble hand-crafted templates. Moreover, the variational autoencoder (VAE) Kingma and Welling (2014) has been extended to produce a disentangled latent space by regularizing the bottleneck layer Higgins et al. (2017). In contrast, we optimize for DNNs with disentangled internal representations of classes without explicit constraints on the loss, modifications to the architecture, or hand-crafted activation patterns. This also allows for the use of non-differentiable objectives.

## M Extended Crossover and Mutation Details

Mating comprises two core operations: crossover and mutation. The *crossover* operator produces offspring by combining the encodings of two parents. The operator combines the building blocks between successful parents to exploit the *implicit parallelism* of population-based search (Holland, 1992). Due to the integer-based encoding that we employ in this work, we elect to use simulated binary crossover (Deb et al., 2007), which uses a probability density function to simulate the single-point crossover of binary-coded genetic algorithms. The *mutation* operator produces offspring

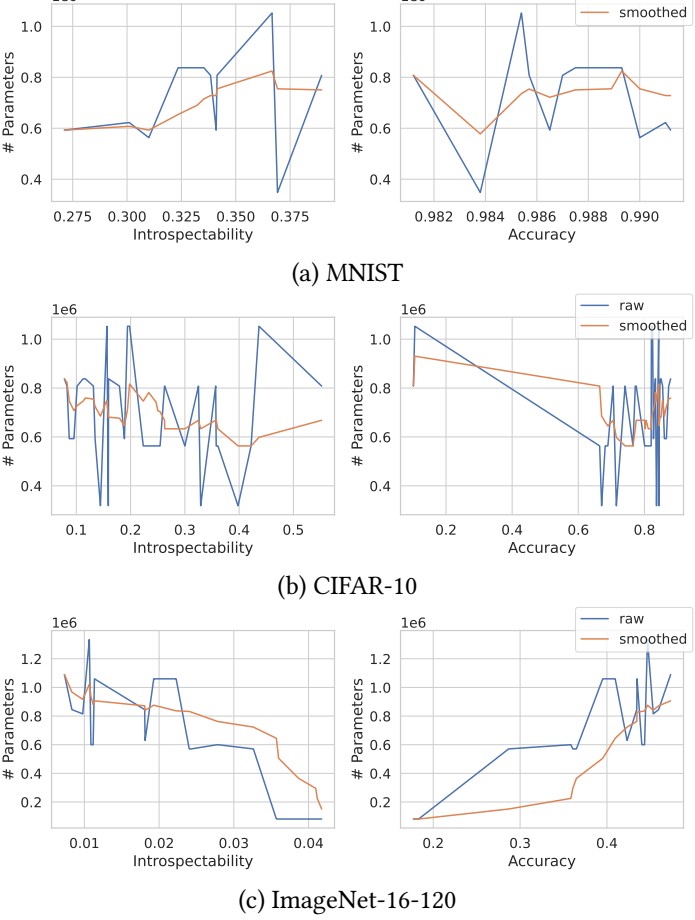

(a) MNIST

(b) CIFAR-10

(c) ImageNet-16-120

Figure J2: The effect of introspectability and accuracy on the number of parameters across the tasks.

by modulating one or more of the variables of a single parent. We specifically select polynomial mutation, which follows the same probability distribution as simulated binary crossover. Both crossover and mutation also have a parameter $p$ that controls the probability that the respective operator is applied to a member of the population.

## N   Dataset Information

**MNIST**. The MNIST LeCun et al. (2010) dataset is available at `https://yann.lecun.com/exdb/mnist/`. Yann LeCun and Corinna Cortes hold the copyright of MNIST dataset, which is a derivative work from the original NIST datasets. MNIST dataset is made available under the terms of the Creative Commons Attribution-Share Alike 3.0 license. The dataset does not contain personally identifiable information or offensive content.

**CIFAR-10**. The CIFAR-10 Krizhevsky (2009) dataset is available at `https://www.cs.toronto.edu/%7Ekriz/cifar.html`. There is no license provided for the dataset. The dataset does not contain personally identifiable information or offensive content.

**ImageNet-16-120**. The ImageNet-16-120 (Dong and Yang, 2020) dataset is a subset of ImageNet which is available at `https://www.image-net.org/download.php`. The terms of using the ImageNet dataset are also outlined at `https://www.image-net.org/download.php`. ImageNet does not own the copyright to the images, rather it compiles a list of web images per synset as described at `https:`

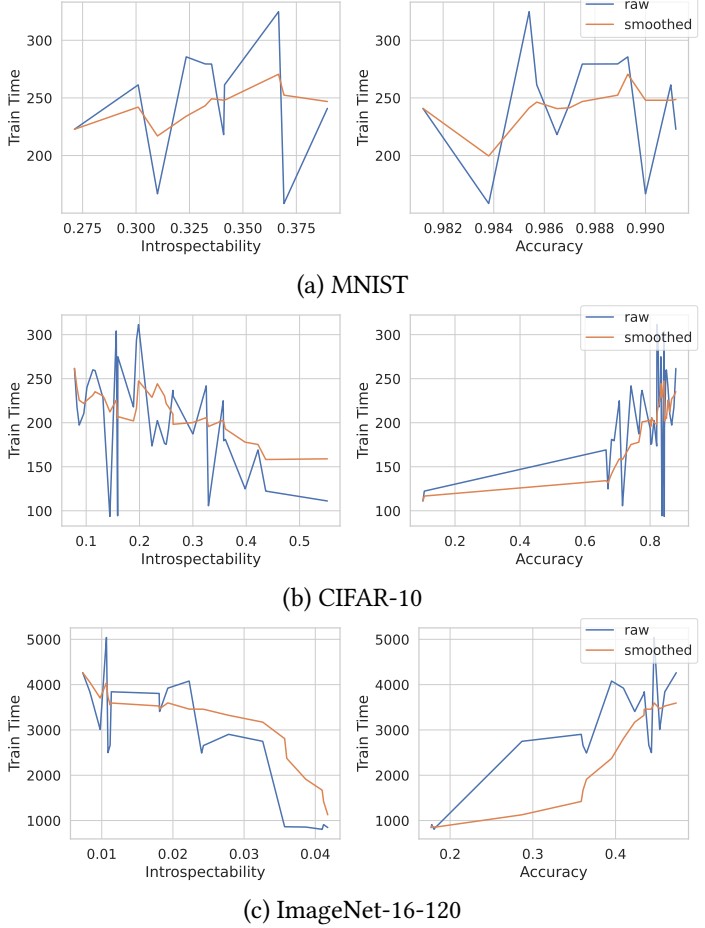

(a) MNIST

(b) CIFAR-10

(c) ImageNet-16-120

Figure J3: The effect of introspectability and accuracy on the training speed across the tasks.

`//www.image-net.org/about.php`. The dataset is known to contain some personally identifiable information and potentially offensive content – see Asano et al. (2021) for further details.

## O   Scaling Up XNAS

We scale XNAS to clusters comprising an arbitrary number of compute nodes using the distributed framework, Ray (Moritz et al., 2018). Given a set of $M$ nodes $\{n_i\}_{i=1}^{M}$, $n_1$ is treated as a head node that is responsible for running the core NSGA-II optimization loop and the core Ray server. The remaining nodes $\{n_i\}_{i=2}^{M}$ are configured as workers available to train and evaluate architectures on a dataset. When a new generation of architectures is created, each offspring is submitted for fitness evaluation to a queue by the head node. Each job in the queue is offloaded to a free worker until all workers complete their jobs and the queue is empty. The head node $n_1$ also is treated as an additional worker if it has free resources. Each worker node can execute in parallel as many jobs as it has GPUs.

## P   FLOPs Analysis

Here, we show the relationship between FLOPs and the two objectives, accuracy and introspectability. While fewer convolutional layers are preferred by the introspectability metric, there is not a direct relationship between either objective and FLOPs. Figure P1 shows the relationship for the

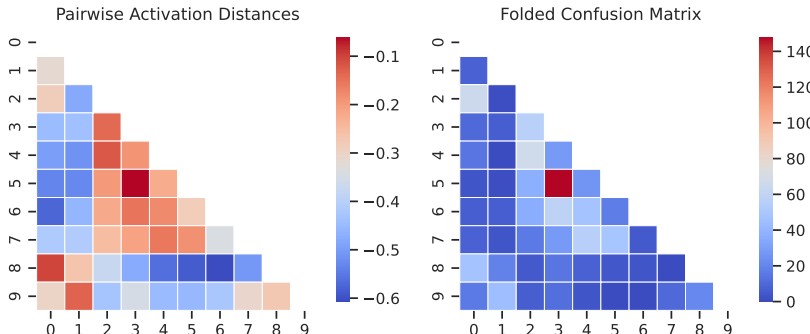

Figure K1: Heat maps of (left) the negated pairwise activation distances as part of the introspectability computation, and (right) the ground truth confusion matrix folded along the diagonal. The heat maps are shown for a random model trained on CIFAR-10. As can be seen, the model confounds the classes 3 and 5, which is reflected in the distance between activations.

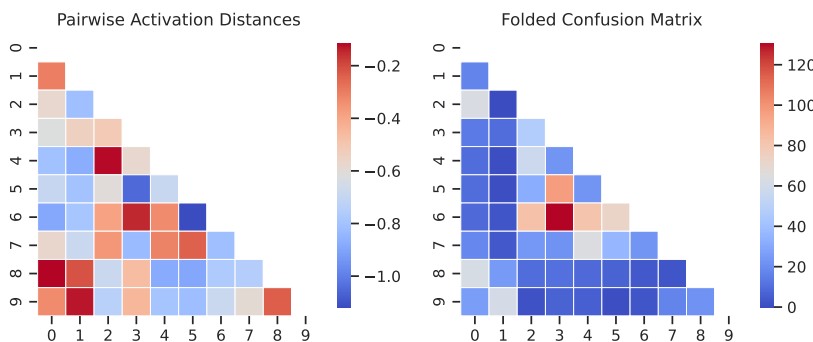

Figure K2: Heat maps after the correction procedure is run of (left) the negated pairwise activation distances as part of the introspectability computation, and (right) the ground truth confusion matrix folded along the diagonal. Note the difference in color scale from Figure K1.

MNIST, CIFAR-10, and ImageNet-16-120 datasets for the Pareto fronts discovered by multi-objective XNAS.

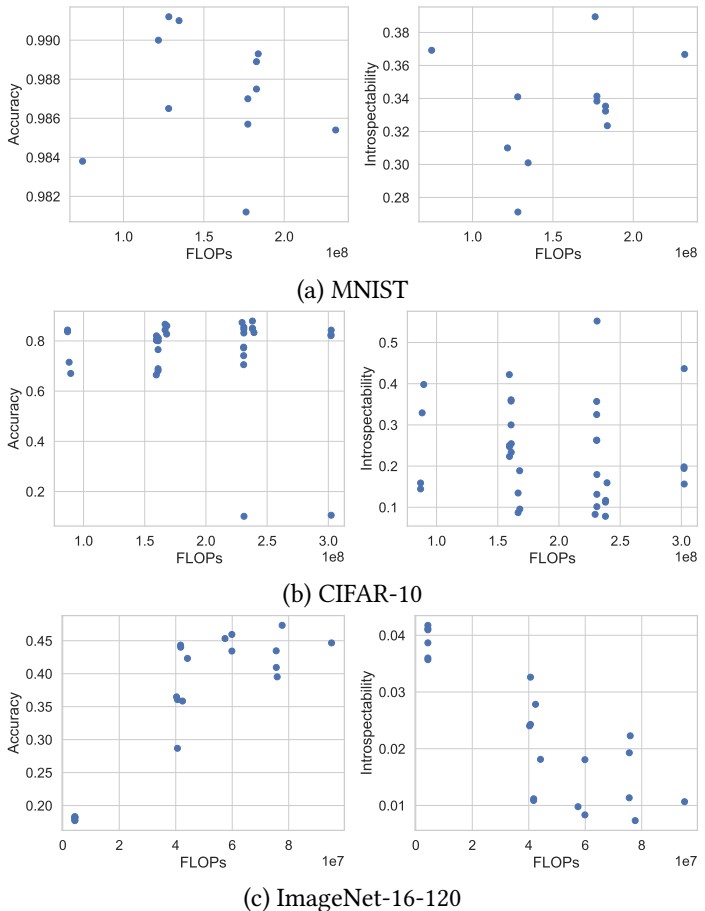

(a) MNIST

(b) CIFAR-10

(c) ImageNet-16-120

Figure P1: Relationship between FLOPs and the two objectives, accuracy and introspectability, on the MNIST, CIFAR-10, and ImageNet-16-120 datasets for the Pareto fronts discovered by multi-objective XNAS.

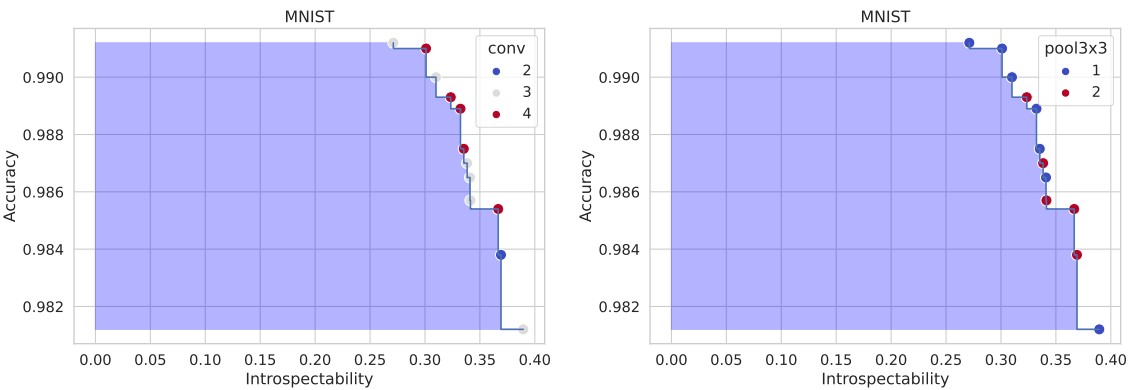

Figure Q1: The Pareto front for the MNIST task with solutions colored by the number of convolutional (left) and pooling (right) layers.

# Q  Motif Figures

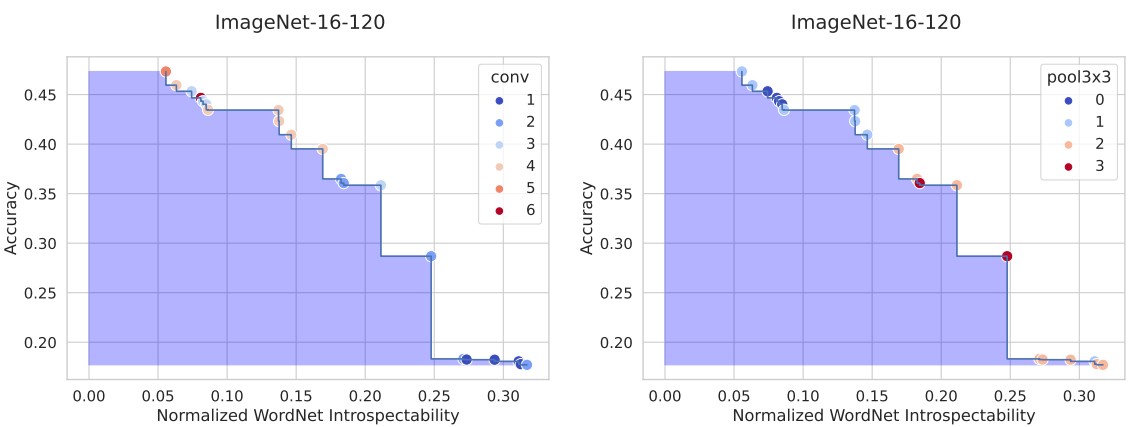

Figure Q2: The Pareto front for the ImageNet-16-120 task with solutions colored by the number of convolutional (left) and pooling (right) layers.

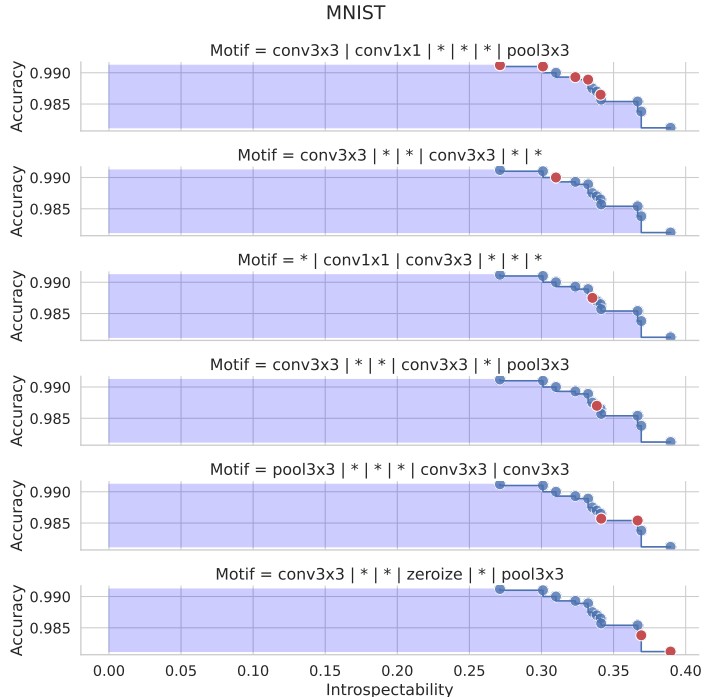

Figure Q3: All discovered motifs among the Pareto optimal solutions on the MNIST task. See text for description of the motif discovery process. Each red solution indicates that its architecture has the motif shown in the sub-plot title. The remaining solutions are shown in blue. For the N/A plot, none of the discovered motifs apply to the architecture.

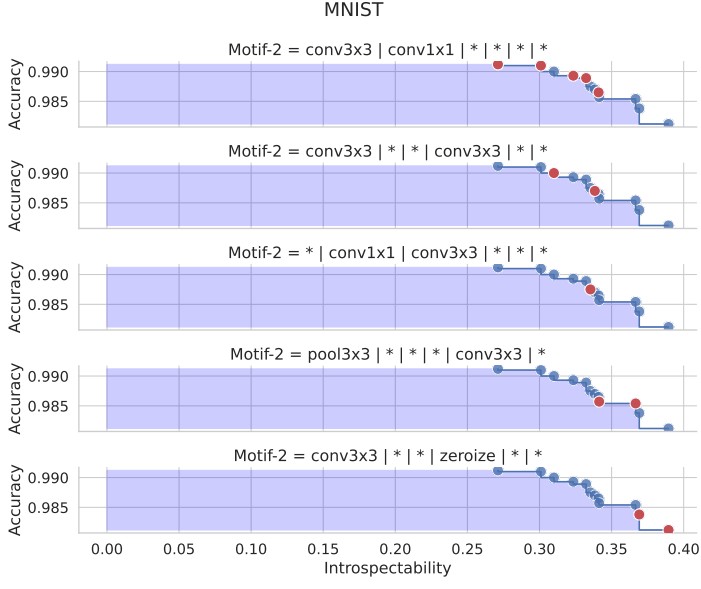

Figure Q4: Discovered motifs of size 2 among the Pareto optimal solutions on the MNIST task. See text for description of the motif discovery process. Each red solution indicates that its architecture has the motif shown in the sub-plot title. The remaining solutions are shown in blue. For the N/A plot, none of the discovered motifs apply to the architecture.

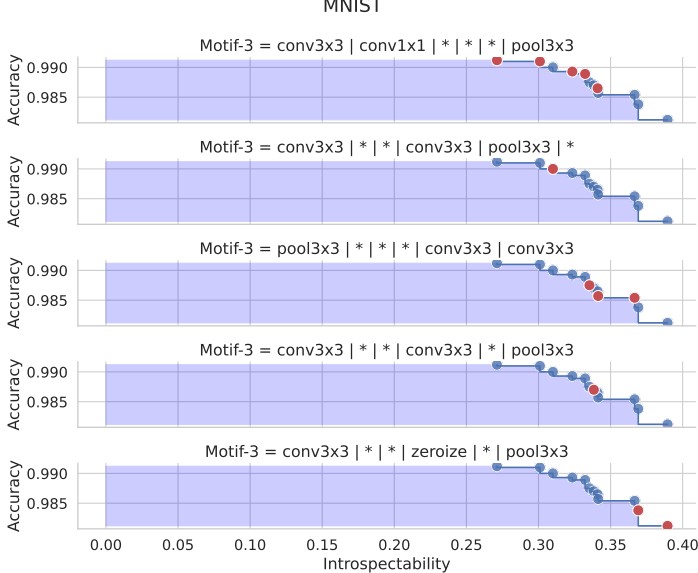

Figure Q5: Discovered motifs of size 3 among the Pareto optimal solutions on the MNIST task. See text for description of the motif discovery process. Each red solution indicates that its architecture has the motif shown in the sub-plot title. The remaining solutions are shown in blue. For the N/A plot, none of the discovered motifs apply to the architecture.

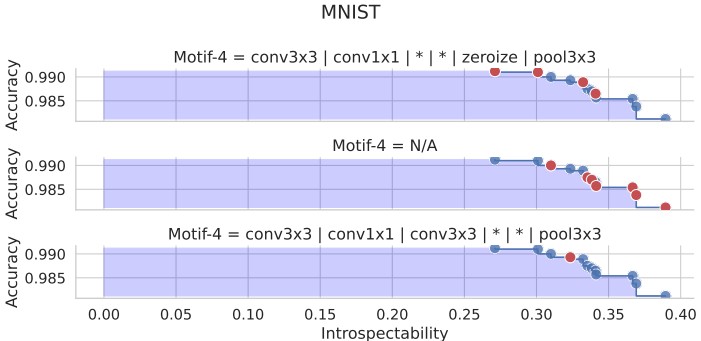

Figure Q6: Discovered motifs of size 4 among the Pareto optimal solutions on the MNIST task. See text for description of the motif discovery process. Each red solution indicates that its architecture has the motif shown in the sub-plot title. The remaining solutions are shown in blue. For the N/A plot, none of the discovered motifs apply to the architecture.

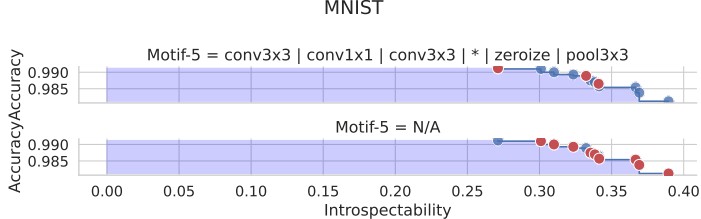

Figure Q7: Discovered motifs of size 5 among the Pareto optimal solutions on the MNIST task. See text for description of the motif discovery process. Each red solution indicates that its architecture has the motif shown in the sub-plot title. The remaining solutions are shown in blue. For the N/A plot, none of the discovered motifs apply to the architecture.

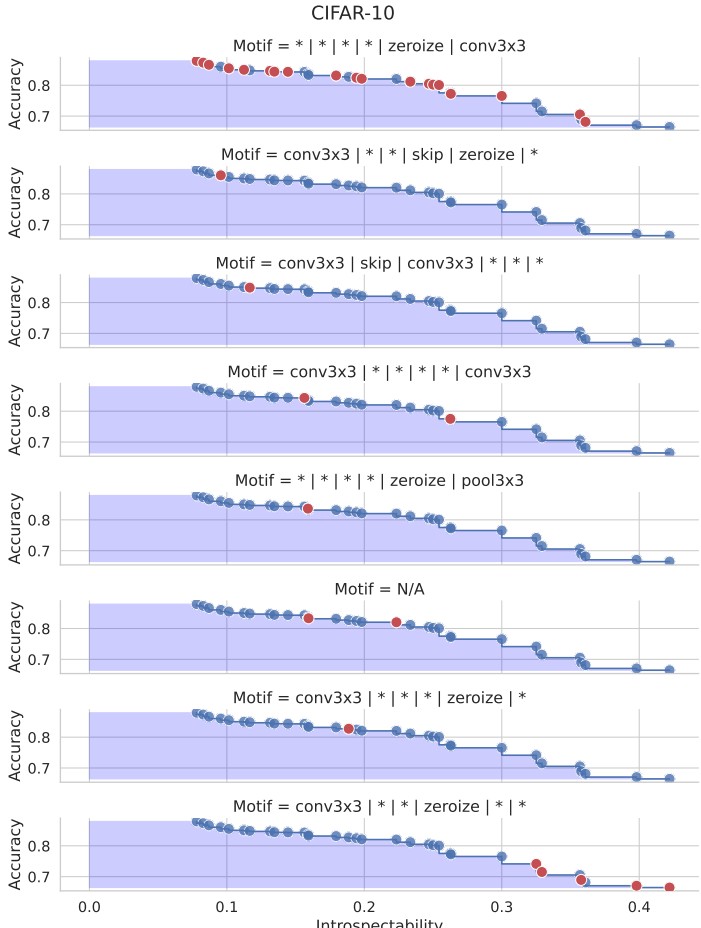

Figure Q8: All discovered motifs among the Pareto optimal solutions on the CIFAR-10 task. See text for description of the motif discovery process. Each red solution indicates that its architecture has the motif shown in the sub-plot title. The remaining solutions are shown in blue. For the N/A plot, none of the discovered motifs apply to the architecture.

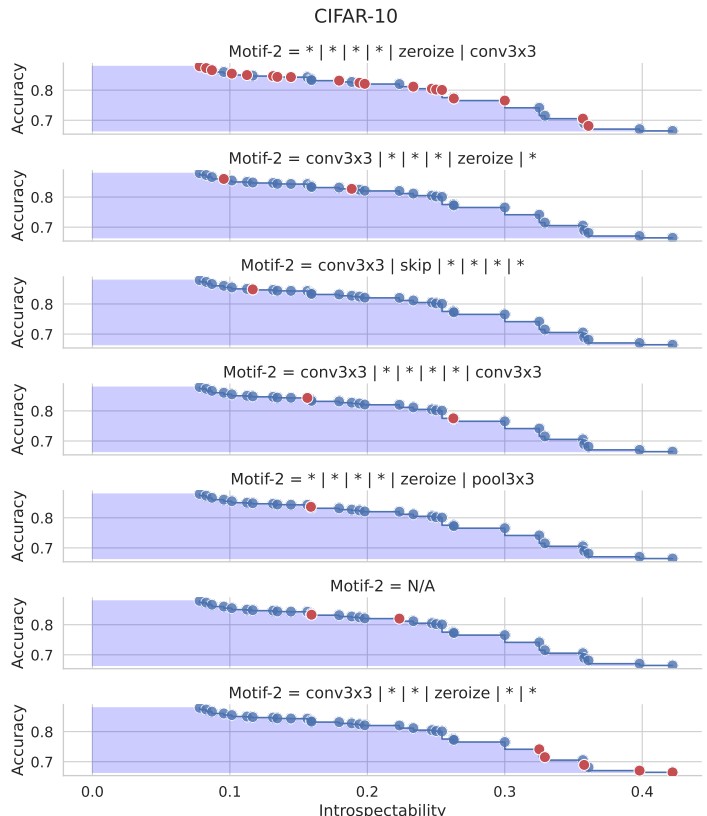

Figure Q9: Discovered motifs of size 2 among the Pareto optimal solutions on the CIFAR-10 task. See text for description of the motif discovery process. Each red solution indicates that its architecture has the motif shown in the sub-plot title. The remaining solutions are shown in blue. For the N/A plot, none of the discovered motifs apply to the architecture.

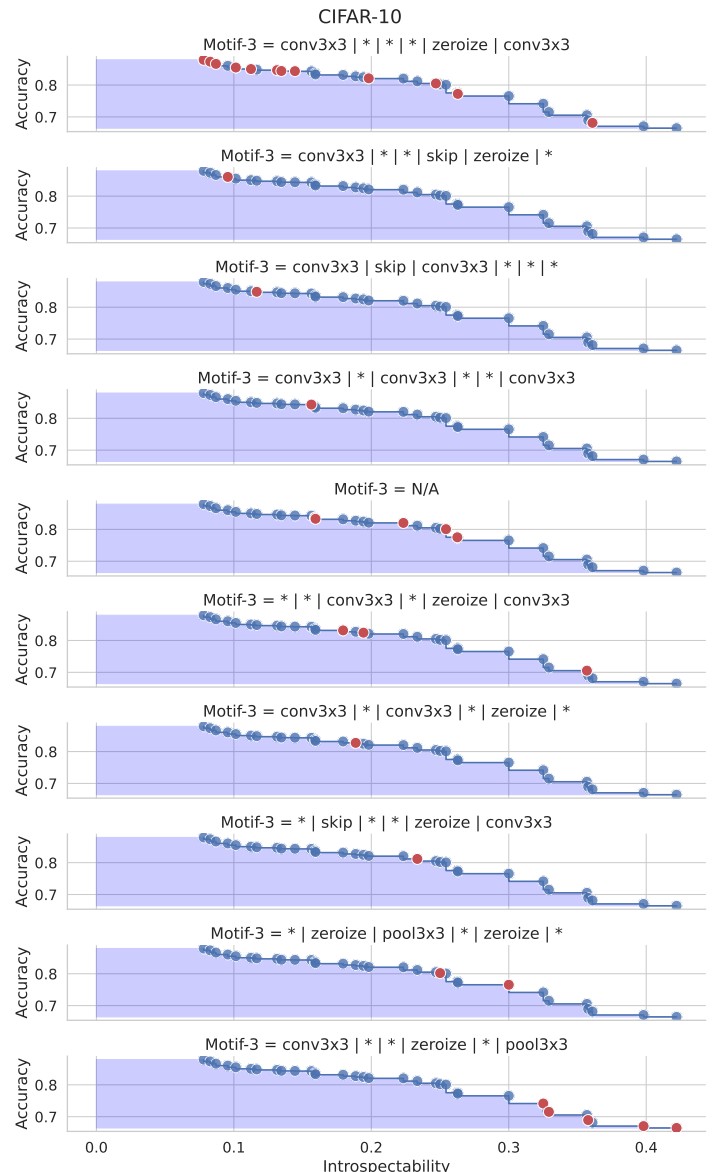

Figure Q10: Discovered motifs of size 3 among the Pareto optimal solutions on the CIFAR-10 task. See text for description of the motif discovery process. Each red solution indicates that its architecture has the motif shown in the sub-plot title. The remaining solutions are shown in blue. For the N/A plot, none of the discovered motifs apply to the architecture.

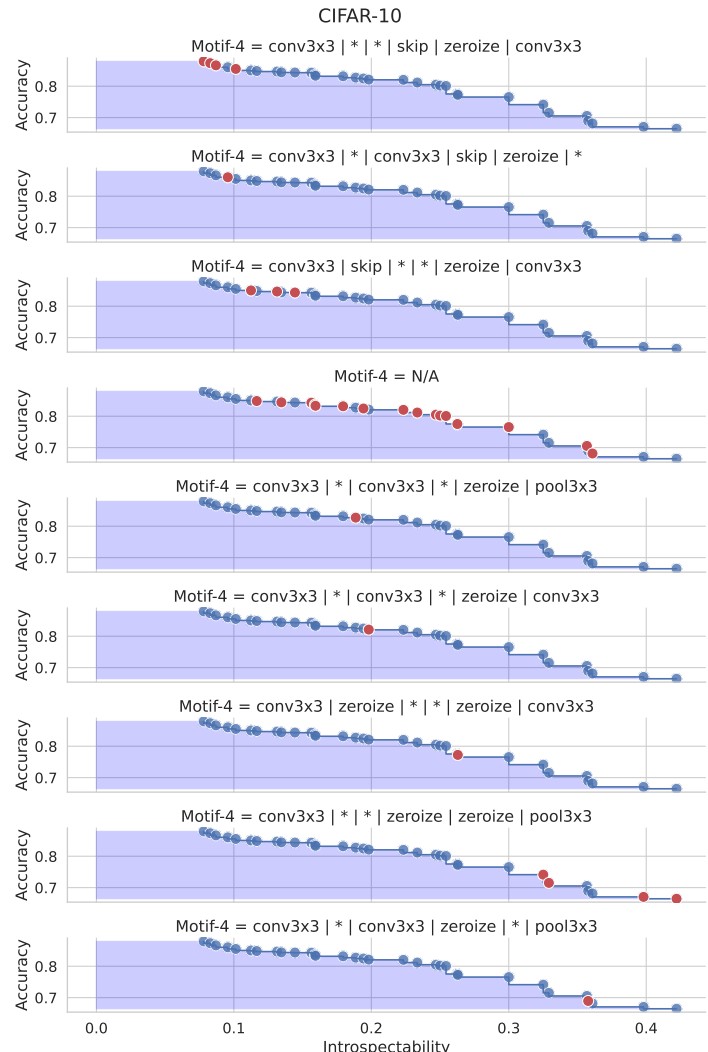

Figure Q11: Discovered motifs of size 4 among the Pareto optimal solutions on the CIFAR-10 task. See text for description of the motif discovery process. Each red solution indicates that its architecture has the motif shown in the sub-plot title. The remaining solutions are shown in blue. For the N/A plot, none of the discovered motifs apply to the architecture.

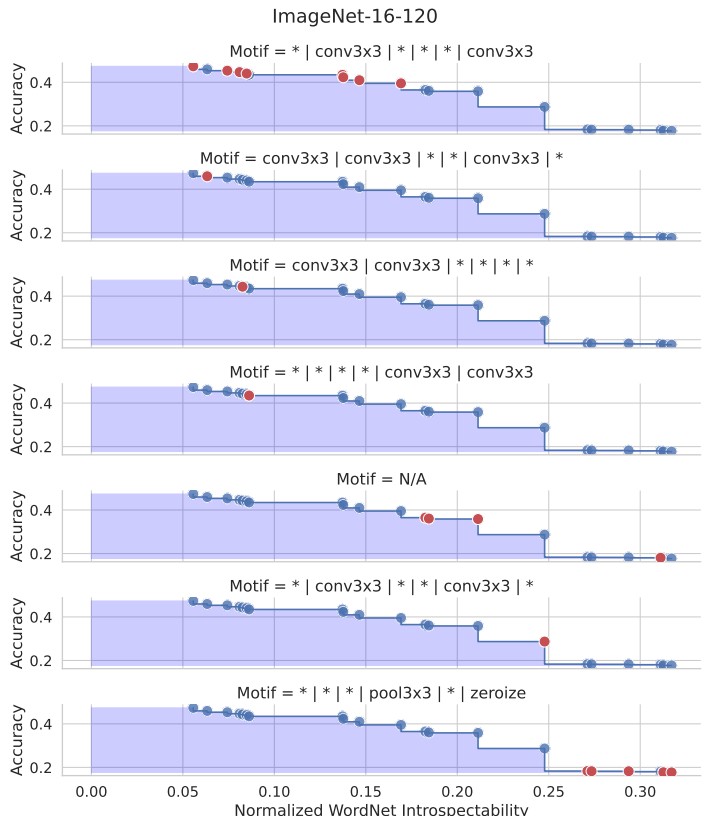

Figure Q12: All discovered motifs among the Pareto optimal solutions on the ImageNet-16-120 task. See text for description of the motif discovery process. Each red solution indicates that its architecture has the motif shown in the sub-plot title. The remaining solutions are shown in blue. For the N/A plot, none of the discovered motifs apply to the architecture.

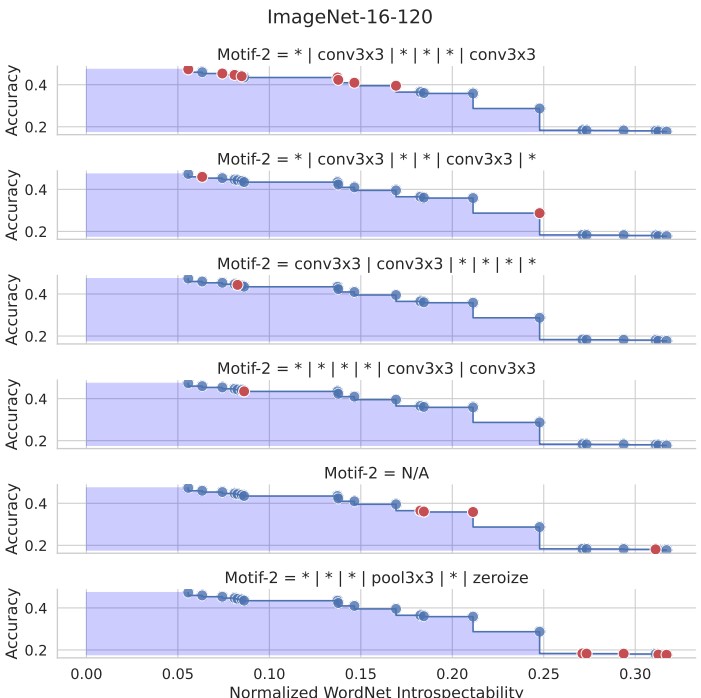

Figure Q13: Discovered motifs of size 2 among the Pareto optimal solutions on the ImageNet-16-120 task. See text for description of the motif discovery process. Each red solution indicates that its architecture has the motif shown in the sub-plot title. The remaining solutions are shown in blue. For the N/A plot, none of the discovered motifs apply to the architecture.

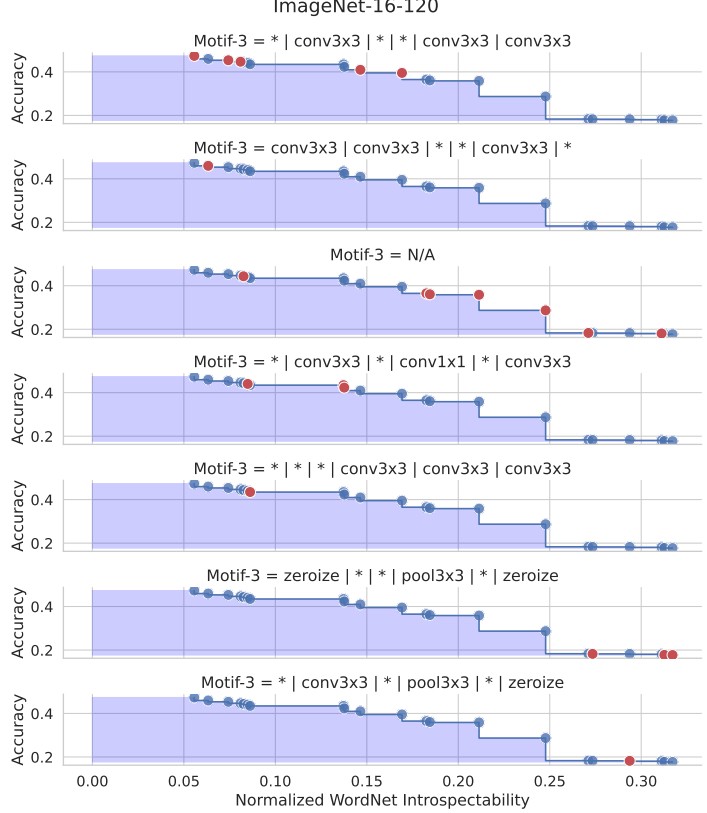

Figure Q14: Discovered motifs of size 3 among the Pareto optimal solutions on the ImageNet-16-120 task. See text for description of the motif discovery process. Each red solution indicates that its architecture has the motif shown in the sub-plot title. The remaining solutions are shown in blue. For the N/A plot, none of the discovered motifs apply to the architecture.

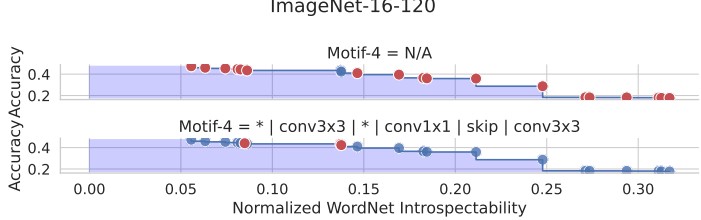

Figure Q15: Discovered motifs of size 4 among the Pareto optimal solutions on the ImageNet-16-120 task. See text for description of the motif discovery process. Each red solution indicates that its architecture has the motif shown in the sub-plot title. The remaining solutions are shown in blue. For the N/A plot, none of the discovered motifs apply to the architecture.

