# OpenReview forum: "Learning Debuggable Models Through Multi-Objective NAS"
_automl.cc/AutoML/2023/Conference — AutoML 2023 Workshop_

### Review · Reproducibility_Reviewer_LR3P · 2023-04-11

**Completeness Of Code And Dataset Supplement Rating:** 3
**Usability And Ease Of Reproducibility Rating:** 3

**Actions Required To Increase The Reproducibility And Overall Recommendation:**

I would suggest the following improvements:
    - Be more specific in the checklist regard where to find the answers to the questions inside the supplementary material.
    - Add more information about how to reproduce the plots
    - Add more information about "how to run the XNAS experiments using the interpretability as regularization, single objective and multi-objective approach followed by fine-tuning with the regularizer (“✓+reg”"
    - Fix the requirements.txt or simply remove it and force users to use conda only.
    - Fix import "from xnas.nas_deephyper.regevo_xnas" to "regevo_bnas"


**Completeness Of Code And Dataset Supplement:**

The README.md does not include information about the plots and neither any other place that i could find. This is the weakest part of the code documentation. On the other hand the main code to train the networks and gather the results has reasonable information and contains all the necessary code to reproduce the proposed methods results.

**Overall Reproducibility Review:**

Positive:
  - Reuse lots of available tools to create a new symbiotic and easy-to-use script/tools.
  - Code is well organized, structured and clean.
  - The authors provide raw results in order to make reproduction and inspection easy

Negative
  - lack of documentation in the code and inside README.md realated to the plots.
  - Missing code execution parameters about "how to run the XNAS experiments using the interpretability as regularization, single objective and multi-objective approach followed by fine-tuning with the regularizer (“✓+reg”"

**Review Confidence:**

4: You are confident in your assessment, but not absolutely certain. It is unlikely, but not impossible, that you did not understand some parts of the submission or that you are unfamiliar with some pieces of the code or data.

**Review Rating:**

8: Accept, all aspects of this are reproducible with minor effort.

**Review Summary:**

The authors made a great code in this paper. They were able to reuse lot of great available tools and merge then into a functional method for NAS. Although some parts need more documentation, it should not be hard to them to describe it a little bit more given the good code organization.

**Summary Of Necessary Code And Dataset Supplement:**

The authors used many tools together in order to create their method. They used DeepHyper to loop over the NAS training with NSGA-II (from pymoo) to explore the NAS-Bench-201 search space with MNIST, CIFAR10 and ImageNet Datasets.
They also had to implement introspectability metric and also implemented how to weight it compared to the accuracy metric.
It is also important to highlight that they boldly extended DeepHyper NeuralArchitectureSearch class and used its Distributed computing support.

**Usability And Ease Of Reproducibility:**

I was to train some models using MNIST but not for all dataset given the lack of time (I lost a lot of time setting up Tensorflow and the version they required was no compatible with my GPU driver).
I was able to train many MNIST models, but the visualization (mainly the Pareto frontier) was not well documented so i was not able to plot it. Even though, I truly believe that a research with more time would be able to reproduce they results with the current code.
Also there is one important improvement that requires additional documentation: It is not clear in the README.md as well as in the supplementary material how to run the XNAS experiments using the interpretability as regularization, single objective and multi-objective approach followed by fine-tuning with the regularizer (“✓+reg”).

Here are some other findings while trying to run it:

Dependencies conflict in the requirements.txt file
The conflict is caused by:
    The user requested numpy==1.21.1
    scipy 1.7.1 depends on numpy<1.23.0 and >=1.16.5
    pandas 1.3.1 depends on numpy>=1.17.3
    tensorflow-gpu 2.4.1 depends on numpy~=1.19.2

fixed using
numpy~=1.19.2
Had to add tensorflow otherwise all versions was being tested for more than an hour
tensorflow==2.4.1
finally i had to add scikit-learn to fix more conflicts
scikit-learn==0.24.2

from xnas.nas_deephyper.regevo_bnas was using the wrong file name (regevo_xnas)

It was necessary to start "ray" locally before running the ./deephyper_xnas main entry point

Overall there is a lot of more information needed to run tensorflow. Although this is not a responsibility of the paper per se, it would be nice to mention that additional steps are need and point to some sources.

---

> ### Author Response · Authors · 2023-05-01
> **Author Response to Reviewer LR3P**
>
> Thanks so much for the detailed and constructive feedback! Based on your great feedback, we have made some changes to the code. We address individual concerns in the following:
>
> > The README.md does not include information about the plots and neither any other place that i could find. This is the weakest part of the code documentation.
>
> We have updated README.md to document this information in the subsection "Processing Raw Results."
>
> > Overall writing solely "See the supplemental material." in the checklist whereas the supplementary material is 32 pages long is a waste of argumentative and descriptive space that makes both author and review work harder.
>
> We sincerely apologize for not being as concise here and making you jump through hoops to find this information. It was not our intent and a lapse in our foresight. While it may not help you now, we have updated the checklist to point to specific sections in the supplemental material.
>
> > I disagree [with...] (c) [...] I was not able to find it, can you please be more specific? [...] (d) [...] The code is organized and structured in a good fashion, but there is no comments or docstrings (i.e. documentation) [...] (l) [...] The correct answer is YES, because they used NAS-Bench-201. Please refer to https://jmlr.org/papers/volume21/20-056/20-056.pdf, best practice 10.
>
> We have updated our responses to the concerns raised with (c), (d), and (l) in the paper. To summarize, we added the code reference and pointer to its documentation, discussed why code documentation was removed in the submission (the final version will not have comments stripped), and corrected our misunderstanding with (l).
>
> > Missing code execution parameters about "how to run the XNAS experiments using the interpretability as regularization, single objective and multi-objective approach followed by fine-tuning with the regularizer (“✓+reg”"
>
> We have updated README.md to discuss how to run this approach with the code base - see "Running Introspectability as a Regularizer."
>
> > Add more information about how to reproduce the plots
>
> We have updated README.md to describe how to run the plotting code given the raw data in the subsection "Processing Raw Results."
>
> > Fix the requirements.txt or simply remove it and force users to use conda only.
>
> We have elected to toss `requirements.txt` and support conda only. We're sorry you had so many dependency issues!
>
> > Fix import "from xnas.nas_deephyper.regevo_xnas" to "regevo_bnas"
>
> We have made this fix in the codebase, good catch! This was due to a find+replace before submission that affected the code contents, but not the filenames.
>
> We thank you again for your great feedback and your support of this work.

---

### Official Review · Reviewer_mmgM · 2023-04-12

**Potential Impact On The Field Of Automl Rating:** 2
**Technical Quality And Correctness:** 1. The algorithm of multi-objective N…
**Technical Quality And Correctness Rating:** 2
**Clarity Rating:** 3

**Summary Of Contributions:**

The motivation of this paper is to optimize task performance and interpretability jointly.  This paper proposed an interesting metric, introspectability, to qualify the model's interpretability. Introspectability measures cosine similarity between activation vectors. It is not clear why introspectability can be used to qualify interpretability.  The well-known multiobjective evolutionary algorithm NSGA-II is used to optimize two objectives. The algorithm is demonstrated in NAS-Bench-201 on MNIST, CIFAR10, and ImageNet-16-120.

**Actions Required To Increase Overall Recommendation:**

I raised concerns about introspectability. If the authors can better explain it and provide some toy examples, I'd like to increase the score.

**Clarity:**

DAGs in NAS-Bench-201 are represented as vectors. Each element represents an operator. But I am not sure how you represent connections between nodes. Can you explain it?

**Overall Review:**

This paper provides a new application of multi-objective NAS. But the algorithm contribution is weak. The model's interpretability metric, introspectability, does not make sense to me. I hope the authors can clarify it.

**Potential Impact On The Field Of Automl:**

I feel this work will not bring a lot of discussion to the AutoML community. Because the multi-objective NAS using NSGA-II is well known. It still has the potential to generate discussion of model interpretability.

**Reproducibility (Optional):**

The code is provided. Although I did not run the code, I feel confident about its reproducibility.

**Review Confidence:**

3: You are fairly confident in your assessment. It is possible that you did not understand some parts of the submission or that you are unfamiliar with some pieces of related work.

**Review Rating:**

5: Borderline Leaning Reject: Technically sound paper where reasons to reject nonetheless outweigh reasons to accept. Please use sparingly.

**Review Summary:**

This paper proposed a new application of multi-objective NAS. Algorithms are not novel. How to qualify the model's interpretability is an interesting point. But I have concerns about it as I mentioned. So, my initial rating is borderline reject. I keep some space to adjust the score if the authors can solve my concern. Because the multi-objective NAS part is not new. I expect strong contributions from other sides, ie. model interpretability.

---

> ### Author Response · Authors · 2023-05-01
> **Author Response to Reviewer mmgM (1/2)**
>
> Thank you for your support of our work! Based on your great feedback, we have made some changes to improve the quality of the paper. We address individual concerns in the following:
>
> > It is not clear why introspectability can be used to qualify interpretability. I did not understand why the cosine distance between activation vectors can be used to quantify the model's interpretability. I did not figure out the connections between introspectability and interpretability. Can you provide a toy example to show the connection between introspectability and interoperability? [...] I raised concerns about introspectability. If the authors can better explain it and provide some toy examples, I'd like to increase the score.
>
> We have added new paragraphs in the "Introspectability" section within Section 3.2 that better motivates introspectability and explains how it can be used. Introspectability enables us to have a prediction process that reflects the uncertainty of an instantiated model, a means of probing why decisions were made, and a means of identifying or correcting mispredictions. The mean activations per class can be thought of as the centroids that live in latent space, and the distance from each centroid can be thought of as the likelihood that a prediction is correct (with lower values being more likely). This enables us to identify mispredictions and mislabeled data, to identify why mispredictions happen (e.g., due to similar latent representations between classes), and to correct models under-performing on certain classes (e.g., by explicitly emphasizing latent separation between two confounded classes).
>
> Toy example: Say that we have a model that predicts whether the weather will be sunny (0), rainy (1), or snowy (2). In a model that perfectly maximizes introspectability, the "centroids" corresponding to each class would be pointing in opposite directions. However, since this is not possible for multiple classes, the distances just need to be sufficiently far. For instance, for the toy centroids $a$=[1 0 0], $b$=[0 1 0], and $c$=[0 0 1] we have pairwise distances $ab$=0, $ac$=1, and $bc$=1. These distances inform us how well-separated the classes are in the model, potential interdependence between classes, and enable a variety of debugging. For a queried point, let's say it has the latent representation $d$=[0.5 0.1 0.1] in the model. This has distances of 0.04 to $a$, 0.81 to $b$, and 0.81 to $c$. There are several things we can do with this information. To measure confidence, we can find which centroid that $d$ is closest to ($a$ in this case). The distance to this centroid can be interpreted as inverse likelihood, which we discuss in the experiments section. Alternatively, we can say that since $d$ is significantly close to $a$ (sunny), and say that its ground truth label is rainy, then we can say that the data point is likely mislabeled (with the likelihood being proportional to the distance). Further technical details are given in the experiments section of this example and additional debugging applications. Please let us know if this helps!
>
> > The algorithm of multi-objective NAS is not novel
>
> We apologize if it seemed that we claimed that the underlying NAS algorithm is novel. We use NSGA-II as part of our pipeline and for simplicity we refer to it with the introspectability metric as "XNAS." We have updated the paper, specifically the contributions enumerated in the introduction, to make it clear that the novelty comes from the contributed metric, its use, and subsequent analyses.
>
> > On the other hand, can you provide FLOPs of your searched models? I want to check if introspectability will prefer smaller networks.
>
> You are actually on to something here - introspectability prefers networks with more pooling layers than convolutional layers, which can be confirmed in the main text in Figure 4. As pooling layers are favorable for the metric, the number of parameters is reduced and the number of FLOPs is reduced. However, it is not a direct relationship between FLOPs and introspectability. We have just added figures showing the relationship between FLOPs and the two objectives (accuracy and introspectability) in Appendix P. We also recommend looking at the discovered motifs in Appendices F, G, and Q and how they relate to the two objectives.
>
> (see reply for continued response)

---

> > ### Author Response · Authors · 2023-05-01
> > **Author Response to Reviewer mmgM (2/2)**
> >
> > > In experiments, classification accuracy is not good. That does not make sense to me. Multi-objective optimization will not prevent you from finding extreme solutions, namely high-accuracy solutions.
> >
> > This is a good observation, however, the limited accuracy is not reflective of our approach. Rather, it is limited by the possible architectures within the NAS-Bench-201 search space. These accuracies are actually in the ballpark for the search space - see the reported NAS-Bench-201 accuracies from the original paper for both ImageNet-16-120 and CIFAR-10. Our reported maximum accuracies should be considered here as there is a trade-off with introspectability, as discussed in the paper. For instance, taking a look at ImageNet-16-120, our achieved performance is right at the optimal solutions.
> >
> > > The definition of domination has a problem. Not all items of a are necessarily bigger than b.
> >
> > That is true - not all items of $\mathbf{a}$ need to be bigger than those of $\mathbf{b}$ for a solution to be non-dominated. However, this is the definition of strict domination, which we use to help define the set of non-dominated solutions. If we look at the definition again, $\\{\mathbf{y}^{\prime} \in Y \mid \\{\mathbf{y} \in Y \mid \mathbf{y} \succ \mathbf{y}^{\prime}\\} = \emptyset \\}$, it actually says we want the set of all solutions that are _not_ strictly dominated by another solution. This leaves us with just the non-dominated solutions. Let us know if you are still unclear.
> >
> > > In the literature review of NAS, mentioned works are a little bit old. I hope you can revise this part and add more recent works.
> >
> > We have updated the references in the NAS background sections to include more recent papers, including:
> >
> > - Lopes, Vasco, et al. "Efficient guided evolution for neural architecture search." Proceedings of the Genetic and Evolutionary Computation Conference Companion. 2022.
> > - Baymurzina, Dilyara, et al. "A review of neural architecture search." Neurocomputing 474 (2022): 82-93.
> > - Tan, Mingxing, et al. "Mnasnet: Platform-aware neural architecture search for mobile." Proceedings of the IEEE/CVF conference on computer vision and pattern recognition. 2019.
> > - Zheng, Xinyue, et al. "Disentangled neural architecture search." 2022 International Joint Conference on Neural Networks (IJCNN). IEEE, 2022.
> >
> > > The authors use random sampling to initialize vectors. I suggest they use the Latin Hypercube Sampling (LHS) sampling method.
> >
> > Thank you for suggesting this, it looks quite promising. We have not heard of this to initialize the population in NAS before and will definitely experiment with this in the near future!
> >
> > > DAGs in NAS-Bench-201 are represented as vectors. Each element represents an operator. But I am not sure how you represent connections between nodes. Can you explain it?
> >
> > Absolutely - each DAG is represented as a 6-dimensional vector, which determines the operations within a searched cell. The searched cell is shown in detail in Figure A1 in Appendix A. The connections are based on a fixed structure with four nodes and six edges. Each operation in the vector corresponds to an edge in the cell depending on its position. The text in Appendix A gives further details - let us know if this clears things up!
> >
> > Having addressed your concerns, we hope that you will consider improving your rating. :)

---

### Official Review · Reviewer_XTyx · 2023-04-13

**Potential Impact On The Field Of Automl Rating:** 2
**Technical Quality And Correctness Rating:** 2
**Clarity:** The paper is well structured and clea…
**Clarity Rating:** 4

**Summary Of Contributions:**

The authors propose multi-objective NAS that optimizes both performance and interpretability. The main insight presented by the authors is quantifying the introspectability as the disentanglement between latent representations for different data classes. Experiments show good trade-off can be achieved between accuracy and interospectability.


**Actions Required To Increase Overall Recommendation:**

Suggest the authors to further discuss the insight behind the metric definition, compare with other existing approaches, and elaborate on how the approach can be applied to other tasks.

**Overall Review:**

Positives:
- The paper is well-written and easy to follow.
- The paper addresses a timely problem, and the high-level idea is promising.

Negatives:
- Technical novelty is limited since the main technical contribution is defining a metric via disentanglement.
- Not clear how feasible it is to extend the metric to beyond classification use cases.
- Evaluation is done on a limited scope.

**Potential Impact On The Field Of Automl:**

The evaluation is done on image classification only and with limited number of datasets.

**Review Confidence:**

3: You are fairly confident in your assessment. It is possible that you did not understand some parts of the submission or that you are unfamiliar with some pieces of related work.

**Review Rating:**

3: Reject: For instance, a paper with technical flaws, weak impact, and/or weak evaluation.

**Review Summary:**

Overall, I believe the authors are exploring the right direction, but the detailed approaches as well as the evaluation can be further enhanced.

**Technical Quality And Correctness:**

The idea of defining a metric to quantify introspectability and using it as a surrogate for the multi-objective NAS framework is interesting and promising. However, the authors do not fully convince me why this disentanglement of dataset classes is the right approach. Also, the authors do not discuss how this metric can be extended beyond image classification use cases.

---

> ### Author Response · Authors · 2023-05-01
> **Author Response to Reviewer XTyx**
>
> Thank you for your review. Based on your feedback, we have made some changes to improve the quality of the paper. We address individual concerns in the following:
>
> > However, the authors do not fully convince me why this disentanglement of dataset classes is the right approach.
>
> We have added a new paragraph in the "Introspectability" section within Section 3.2 that better motivates introspectability. Therein, we discuss how introspectability enables us to have a prediction process that reflects the uncertainty of an instantiated model, a means of probing why decisions were made, and a means of identifying or correcting mispredictions. We then, in our experiments section, demonstrate how this can be accomplished as a result of the disentanglement of classes. This is especially convenient as it requires minimal expert domain knowledge. Please let us know if this clears it up and how we can further improve it.
>
> > Technical novelty is limited since the main technical contribution is defining a metric via disentanglement.
>
> We would also like to say that our contributions also include how to optimize for the Pareto front with existing approaches, rigorous analysis of the architectures spanning the Pareto front (trade-offs in debuggability vs. accuracy, architectural motifs, etc.), and demonstration of how introspectability can be used to produce more trustworthy models in several experiments. Furthermore, the implementation of our framework is nontrivial, which we highlight in the supplemental material PDF and code.
>
> > Not clear how feasible it is to extend the metric to beyond classification use cases. the authors do not discuss how this metric can be extended beyond image classification use cases.
>
> While the evaluation on non-classification tasks is outside the scope of the paper, we absolutely agree that this is an important consideration. We have added discussion of how this can be done (with domain knowledge or additional data-driven statistics) in the "Introspectability" section within Section 3.2 (last paragraph) - introspectability is applicable to non-classification tasks (e.g., regression) as well.
>
> > compare with other existing approaches
>
> We think that there is not a straightforward way of comparing the introspectability approach to other approaches out there, at least quantitatively. However, if this comment is concerning NAS methods, we consider the comparison of XNAS with other multi-objective population-based NAS algorithms to be out of scope for this work. While we would like to explore incorporating other NAS methods into our framework for evaluation, we consider this work to be a standalone proof of concept.
>
> Having addressed your concerns, we hope that you will consider improving your rating. :)

---

### Official Review · Reviewer_iLB7 · 2023-04-22

**Potential Impact On The Field Of Automl Rating:** 3
**Technical Quality And Correctness Rating:** 4
**Clarity Rating:** 3

**Summary Of Contributions:**

This paper proposes a novel framework for NAS that simultaneously optimizes for performance and “introspectability,” and a multi-objective evolutionary NAS algorithm called XNAS that optimizes over this pareto frontier. The main improvement introduced with this framework is the increased robustness of this kind of multi-objective optimization over traditional accuracy-only optimization. The authors conducted experiments on the multi-objective performance of XNAS on 3 image classification tasks, and performed analysis on the network motifs and model introspectability, trustworthiness, and debuggability.

**Actions Required To Increase Overall Recommendation:**

For figure 2, it might be better to showcase the difference between only the interpretability of the two optimization tasks rather than the hypervolume. This would showcase the models.

Extending the experiments to cover more applicable tasks might include facial recognition tasks, or other “higher-risk” classification tasks.

For the lack of clear motivation (not in the field of research but in the actual workflow for this research), putting in a description of the effectiveness of leveraging the new explainability outputs over the single objective would be beneficial.

**Clarity:**

This paper was written in a clear and concise manner. The experimental setup, including experimental variables and design choices was discussed in detail. The summary, conclusion, background, and impact statement discussed and elaborated on the objectives of this paper in a logical format. There is some lack of clarity in the discussion of results.

**Overall Review:**

Strengths:

This paper includes a well-thought out experimental design that showcases the effectiveness of using the bi-level optimization problem consisting of the model accuracy and class-wise disentanglement using an average cosine distance for all pairwise comparisons (Section 3.2). XNAS utilizes this performance measure to search for architectures that are prone to separation between classes through the NSGA-II NAS algorithm. This framework for multi-objective optimization is shown to be more robust to identifying mispredictions, improving model explainability, and improving hypervolume in this multi-objective task.

Weaknesses:

Showcasing the model’s performance is good, but using the hypervolume as a comparison against the single objective optimization (figure 2) is somewhat misleading. In this case, it might come across as a means of proof for the improvement of the model when all the figure is showing is that the new framework optimizes over a new objective.
Another weakness in this paper is that it does not describe in detail the thought process behind the design of this interpretability objective and its expected use. There is a mention (lines 261-262) that discuss the motivation behind DNN trustworthiness and disentanglement, but there is no clearly described motivation elsewhere.
Figure 3 is very small, and dense. Subsequently, it is difficult to grasp information from it.
One minor point is on the specificity of modality. By focusing on image classification tasks, it showcases a good general use case scenario, but this model would probably aim to see production into tasks where the higher prediction confidence or trustworthiness is useful (i.e. higher risk tasks) or the better model debuggability is useful, which is not covered in the 3 tasks.


**Potential Impact On The Field Of Automl:**

The impact of this work has broader reach in NAS and other fields through model explainability and the usefulness of model latent space representation separation. This allows the paper to put forth paths towards more reliable and debuggable models with moderate overhead increases.

**Review Confidence:**

3: You are fairly confident in your assessment. It is possible that you did not understand some parts of the submission or that you are unfamiliar with some pieces of related work.

**Review Rating:**

7: Weak Accept: Technically sound paper with moderate-to-high impact and strong evaluation, with perhaps some minor flaws.

**Review Summary:**

Overall, this paper includes thorough experimentation on image classification using the new framework for multi-objective NAS. While there are still some considerations about the overall motivational design behind this research, it clearly identifies a path to create debuggable models through multi-objective NAS.

**Technical Quality And Correctness:**

This paper had thorough experimental design on image classification, and ablation studies. While limited in scope to just using evolutionary NAS, specifically NSGA-II, on image classification, this design choice is not a majorly limiting factor in the general usability of this framework.

---

> ### Author Response · Authors · 2023-05-01
> **Author Response to Reviewer iLB7**
>
> Thank you for your support of our work! Based on your great feedback, we have made some changes to improve the quality of the paper. We address individual concerns in the following:
>
> > Showcasing the model’s performance is good, but using the hypervolume as a comparison against the single objective optimization (figure 2) is somewhat misleading. In this case, it might come across as a means of proof for the improvement of the model when all the figure is showing is that the new framework optimizes over a new objective. [...] For figure 2, it might be better to showcase the difference between only the interpretability of the two optimization tasks rather than the hypervolume.
>
> Our intent here was to demonstrate that our approach performs better than random (not optimizing for introspectability). To quantify how effective our approach is, we computed the hypervolume and observed measurable improvements. However, we can see how this can be misleading. We have updated Figure 2 to more clearly demonstrate that we care about the differences in introspectability mostly, while also showing that the achieved accuracies are still in the same range.
>
> > [The paper] does not describe in detail the thought process behind the design of this interpretability objective and its expected use. There is a mention (lines 261-262) that discuss the motivation behind DNN trustworthiness and disentanglement, but there is no clearly described motivation elsewhere. [...] For the lack of clear motivation (not in the field of research but in the actual workflow for this research), putting in a description of the effectiveness of leveraging the new explainability outputs over the single objective would be beneficial.
>
> We appreciate you pointing this out as the motivation could certainly be made more clear. We have added a new paragraph in the "Introspectability" section within Section 3.2 that better motivates the use of introspectability. Please let us know if this clears it up and how we can further improve it.
>
> > Figure 3 is very small, and dense. Subsequently, it is difficult to grasp information from it.
>
> Thanks to the increase in the page limit, we were able to increase the size of this figure by 25%. It is much more readable now.
>
> > One minor point is on the specificity of modality. By focusing on image classification tasks, it showcases a good general use case scenario, but this model would probably aim to see production into tasks where the higher prediction confidence or trustworthiness is useful (i.e. higher risk tasks) or the better model debuggability is useful, which is not covered in the 3 tasks. [...] Extending the experiments to cover more applicable tasks might include facial recognition tasks, or other “higher-risk” classification tasks.
>
> We agree that this would be a highly appropriate experiment for the proposed framework. However, given the limited rebuttal timeline, we cannot run this experiment in time to include in the paper and thus leave it to future work.
>
> Having addressed your concerns, we hope that you will consider improving your rating. :)